# Mouse embryonic stem cells can differentiate via multiple paths to the same state

James Alexander Briggs[1†], Victor C Li[1†], Seungkyu Lee[2,3], Clifford J Woolf[2,3], Allon Klein[1]*, Marc W Kirschner[1]*

[1]Department of Systems Biology, Harvard Medical School, Boston, United States; [2]Department of Neurobiology, Harvard Medical School, Boston, United States; [3]FM Kirby Neurobiology Center, Boston Children's Hospital, Boston, United States

**Abstract** In embryonic development, cells differentiate through stereotypical sequences of intermediate states to generate particular mature fates. By contrast, driving differentiation by ectopically expressing terminal transcription factors (direct programming) can generate similar fates by alternative routes. How differentiation in direct programming relates to embryonic differentiation is unclear. We applied single-cell RNA sequencing to compare two motor neuron differentiation protocols: a standard protocol approximating the embryonic lineage, and a direct programming method. Both initially undergo similar early neural commitment. Later, the direct programming path diverges into a novel transitional state rather than following the expected embryonic spinal intermediates. The novel state in direct programming has specific and uncharacteristic gene expression. It forms a loop in gene expression space that converges separately onto the same final motor neuron state as the standard path. Despite their different developmental histories, motor neurons from both protocols structurally, functionally, and transcriptionally resemble motor neurons isolated from embryos.

DOI: https://doi.org/10.7554/eLife.26945.001

*For correspondence:
Allon_Klein@hms.harvard.edu
(AK);
marc@hms.harvard.edu (MWK)

†These authors contributed equally to this work

## Introduction

Embryonic development proceeds through defined intermediate states, such as germ layer intermediates, and lineage-specific progenitors. Intermediates ramify into multiple states over time, and specialize their behaviors, ultimately producing a lineage tree that defines each mature cell type by a particular sequence of intermediates. This was first appreciated through classical lineage tracing and cell ablation studies. These studies showed that specifically labeled intermediate states generate stereotyped sets of downstream cell types, and that these downstream cell types fail to form if an intermediate that is upstream in their lineage is ablated (*Takebayashi et al., 2002*; *Kretzschmar and Watt, 2012*). Furthermore, most mature cell states are not produced by multiple differentiation paths in the embryo; the only known exceptions are rare and viewed as special cases, such as in the neural crest lineage.

In contrast to this rigid and hierarchical process, recent protocols that experimentally directly program cell fate suggest that the exact sequence of intermediates defining a lineage may be more flexible (*Mazzoni et al., 2013*; *Li and Kirschner, 2014*; *Son et al., 2011*; *Vierbuchen et al., 2010*; *Ieda et al., 2010*; *Szabo et al., 2010*; *Cohen and Melton, 2011*; *Zhou and Melton, 2008*; *Treutlein et al., 2016*). These studies reveal that mature cell states can be reached through paths that do not involve activation of the intermediate progenitor genes that are essential in embryos. Mouse embryonic stem cells (mESCs), for example, can be converted into motor neurons (MNs) by a process that involves overexpression of three transcription factors, Ngn2+Isl1+Lhx3 (*Mazzoni et al.,*

*2013*; *Velasco et al., 2017*), and that never expresses the neural progenitor transcription factors Sox1 and Olig2 (*Mazzoni et al., 2013*). mESCs can also be driven rapidly into a terminal muscle phenotype without normal upregulation of intermediate genes such as Pax7 and Myf5, through a combination of cell-cycle inhibition and MyoD overexpression (*Li and Kirschner, 2014*). This plasticity of differentiation extends further, to the interconversion of mature cell states. Fibroblasts can be converted into mature neuron phenotypes (*Vierbuchen et al., 2010*; *Treutlein et al., 2016*), including MNs (*Son et al., 2011*), seemingly without completely dedifferentiating and retracing the embryonic lineage, as indicated by lack of expression of specific core pluripotency (Oct4, Sox2 and Nanog) and neural progenitor genes (Nestin) (*Son et al., 2011*).

Although these direct programming (DP) experiments imply the existence of differentiation paths that differ from those in embryos, much of what actually occurs in these new programs remains mysterious. Does DP bypass normal intermediates by short-circuiting the natural lineage, or does it transition through alternative intermediates (*Figure 1A*)? Does it diverge only briefly to bypass specific early or late states, or does it utilize an entirely distinct path (*Figure 1B*)? And can DP converge fully to the same final state that is produced in embryos despite taking an alternative path, or does it simply approximate that state (*Figure 1C*)? These questions have been challenging to answer in part due to the high degree of heterogeneity in direct programming experiments, where unbiased bulk measurements of global gene expression obscure changes, and also because marker genes allowing the isolation of new but potentially important DP-specific intermediates are not known in advance. Here we aimed to overcome these issues by applying single cell RNA sequencing to compare the gene expression trajectories of DP and growth factor guided differentiation of mESCs into MNs over time. Our core research questions are summarized in *Figure 1*.

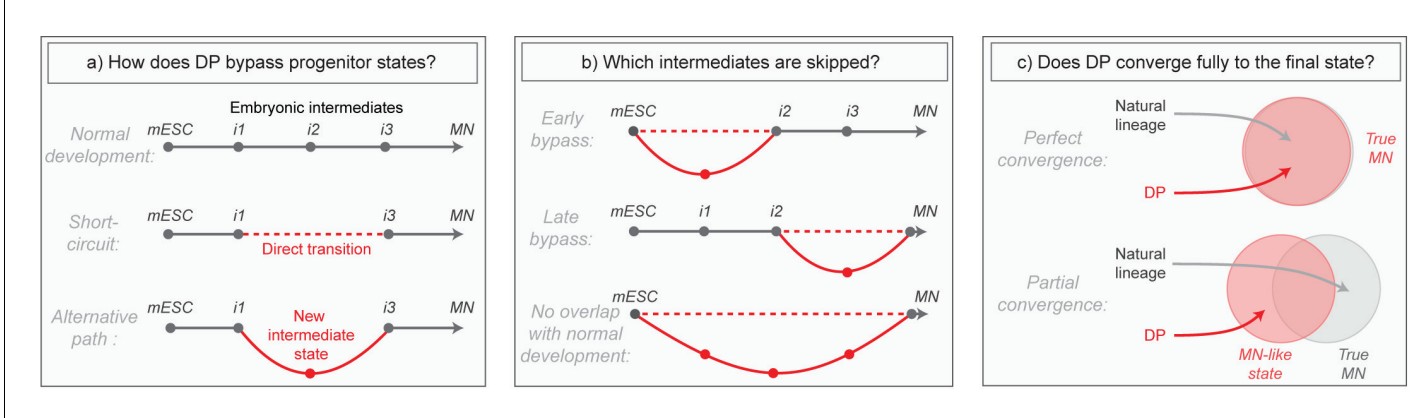

**Figure 1.** Summary of core research questions. (a) Direct programming could in principle skip progenitor states in one of two ways. It may either transition from early states directly into later states through a 'short-circuit' of the natural lineage, or it might utilize an alternative path involving new and potentially abnormal or unstable intermediate states. In the schematic, a conceptual depiction of the natural lineage is shown first, followed by two lineages that bypass the i2 intermediate from the normal lineage, either through short-ciruit or through a new alternative intermediate. (b) DP might diverge from the natural lineage only briefly, or it might access an entirely distinct path. In the schematic, three different possibilities are shown: (i) DP bypasses specific early intermediates, and then converges to the final state through a conserved seqence of terminal cell state transitions; (ii) DP transitions through conserved early stages of differentiation, before diverging into an alternate path that converges separately to the final state; (iii) DP takes an entirely distinct path with no resemblance to the natural lineage. Dashed lines represent a potential direct transition while solid lines represent an alternate path (as shown in the left panel). (c) It is possible that the price paid for taking an alternate differentiation path during direct programming is that cells retain subtle gene expression defects, thus converging only partially to the desired final state. Alternatively, convergence could be near perfect which would imply that differentiation into mature cell states is at least partially history independent. In the schematic, circles are an abstract representation of gene expression state and are used to represent the extent of overlap between MNs generated by either DP or by the natural lineage.

DOI: https://doi.org/10.7554/eLife.26945.002

## Results

### Dissection of two MN differentiation protocols using single cell RNA sequencing

We compared two in vitro differentiation protocols that convert mESCs into spinal MNs, with the goal of identifying and comparing alternate differentiation trajectories between the same start state and similar – if not the same – end states. The first, standard protocol (SP) is a widely used method that attempts to drive cells into a spinal MN state through sequential addition of developmental signals that normally guide MN differentiation in the embryo (Fgfs, Retinoic Acid, and Sonic hedgehog) (*Figure 1A*) (*Wichterle et al., 2002*; *Wu et al., 2012*). Many variations of this method exist, including protocols that use 2D or 3D culture formats, serum or serum-free basal culture medias, and even partially different cocktails of growth factors. Yet, each variant is believed to shuttle cells through the same sequences of intermediate states (*Sances et al., 2016*), namely those that exist in the embryo. As a contrast to SP, we used an alternative direct programming (DP) strategy as our second method. The DP protocol drives cells toward a spinal MN fate by forcing the expression of spinal MN transcription factors (Ngn2+Isl1+Lhx3) (*Mazzoni et al., 2013*; *Velasco et al., 2017*) in cells grown in a simple growth factor free medium (*Li and Kirschner, 2014*). Previous studies have shown that this DP protocol does not induce key marker genes of some embryonic MN intermediate states; for this reason DP was among the first differentiation protocols for which it was explicitly argued that a distinct differentiation process may be occurring (*Mazzoni et al., 2013*; *Velasco et al., 2017*). Our focus in this study was to understand how a non-embryonic differentiation trajectory differs from the SP, which drives cells through an embryo-like sequence of states (*Figure 1*).

We used single-cell RNA sequencing (InDrops) (*Klein et al., 2015*) to profile the differentiation in both protocols over time (*Figure 1B and C*). Single-cell transcriptomics was our method of choice because of its ability to parse cell states and differentiation trajectories within populations that are not pure or contain rare intermediates (*Treutlein et al., 2016*; *Treutlein et al., 2014*; *Scialdone et al., 2016*; *Trapnell et al., 2014*; *Setty et al., 2016*; *Bendall et al., 2014*), as expected for both DP and SP. We initially profiled 4590 single-cell transcriptomes sampled from early (day 4/5) and late (day 11/12) timepoints for each protocol, and also used our previously published data from 975 mES cells as a day 0 reference (*Klein et al., 2015*). To visualize the combined single cell time course data for each protocol, we followed the same procedure described in Klein et al (*Klein et al., 2015*).: we identified highly-variable genes in each data set, retained non-noise dimensions after a principal component analysis on the standardized gene expression scores, and then employed t-distributed stochastic neighbor embedding (tSNE) (*van der Maaten and Hinton, 2008*) to visualize the data (*Figure 2B–C*; further details in Supplementary Methods). The most immediate feature revealed by tSNE was a gene expression continuum for each protocol, which correlated with chronology, suggestive of a differentiation trajectory (*Figure 2B–C*, inset). Several disconnected cell clusters were also visible outside of the main trajectory.

By examining gene expression on the continua and disconnected cell clusters, we annotated the observed cell states produced in DP and SP. We applied unsupervised density gradient clustering (DBSCAN) to each tSNE map to partition cells into clusters, as described in Macosko et al (*Macosko et al., 2015*). This approach robustly partitions each of the disconnected cell clusters, but can also identify boundaries between states within continua, although the latter are less reliably defined. The cells states identified in this manner possessed unique transcriptional signatures, including expression of specific transcription factors, signaling molecules, and structural genes such as synaptic components and neurotransmitters. To annotate each cell state cluster, we identified marker genes for each subpopulation, and used prior knowledge about the characteristics of these genes to interpret the identity of each state. The criteria used for each state are provided in *Figure 2—source datas 1–2*, and the expression of specific marker genes is shown in *Figure 2D–E* and *Figure 2—figure supplements 1–2*.

These cell state annotations reinforced our interpretation of the main gene expression continua in DP and SP as representing differentiation trajectories. In addition to correlating with chronology, the ordering of cell states appeared sensible in both cases. Each trajectory begins with mESCs (Pou5f1+/Esrrb+/Nanog+), passes through familiar neural progenitor states (including for example neural progenitors (NPs) expressing Sox1, Pax6, and Lin28), progresses to early motor neurons (EMNs) (expressing Mnx1, Isl1, Isl2, Nefl, Nefm, Tubb3, and Map2), and terminates at a mature, or late, MN

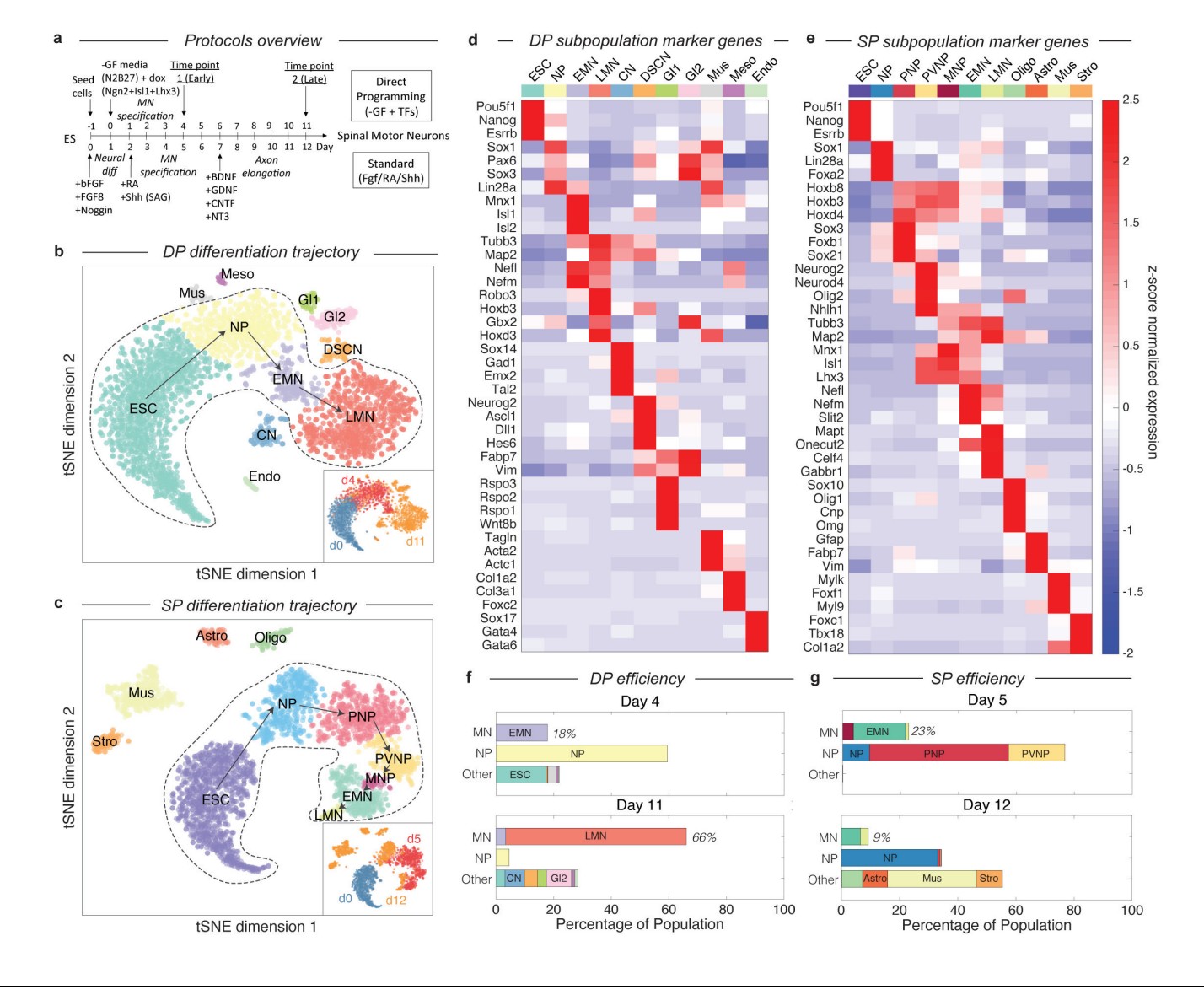

**Figure 2.** Dissection of DP and SP motor neuron differentiation strategies using InDrops single cell RNA sequencing. (a) Summary of the direct programming (DP) and standard protocol (SP) differentiation strategies. (b–c) tSNE visualization of single cell RNA sequencing data from each differentiation strategy. Timepoints are shown inset. Cell state clusters are color coded and annotated with their identities: ESC = embryonic stem cell; NP = neural progenitor; PNP = posterior neural progenitor; PVNP = posterior and ventral neural progenitor; MNP = motor neuron progenitor; EMN = early motor neuron; LMN = late motor neuron; CN = cortical neuron; DSCN = dorsal spinal cord neuron; Gl1 = glia type 1; Gl2 = glia type 2; Mus = muscle; Meso = mesoderm; Endo = endoderm; Oligo = oligodendrocyte; Astro = astrocyte; Stro = stromal. Arrows inside the bounded area indicate the hypothesized cell state progression during MN differentiation for each method. (d–e) Expression heatmap of marker genes used to identify each subpopulation. Colors and annotations for the subpopulations are the same. (f–g) Subpopulation abundances for each protocol over time. DP has significantly higher MN production efficiency than SP (66% vs. 9%) at the late timepoint. MN = EMN or LMN; NP = NP, PNP, or PVNP; Other = everything else. Colors and labels match b-e).

DOI: https://doi.org/10.7554/eLife.26945.003

The following source data and figure supplements are available for figure 2:

**Source data 1.** Summary of criteria used to annotate DP cell states.

DOI: https://doi.org/10.7554/eLife.26945.007

**Source data 2.** Summary of criteria used to annotate SP cell states.

DOI: https://doi.org/10.7554/eLife.26945.008

**Figure supplement 1.** tSNE visualization of marker genes for DP subpopulations.

DOI: https://doi.org/10.7554/eLife.26945.004

*Figure 2 continued on next page*

*Figure 2 continued*

**Figure supplement 2.** tSNE visualization of marker genes for SP subpopulations.
DOI: https://doi.org/10.7554/eLife.26945.005
**Figure supplement 3.** qPCR validation that MN gene expression in DP follows the expected dynamics and does not involve Olig2 induction.
DOI: https://doi.org/10.7554/eLife.26945.006

state (LMN) (downregulating MN progenitor genes Mnx1, Isl1, and Isl2, while upregulating genes indicative of neuronal maturation Robo3, Onecut2, Syn1, Vacht, and Gabr1) (*Figure 2B–E*). Cells along both trajectories overlap between consecutive time points, which would be expected due to asynchrony in differentiation. This was reassuring to us that our experiments sampled all intermediate states. Below we define and compare the specific cell-state transitions that occur during MN differentiation in DP and SP, after first examining the efficiency of MN generation in each protocol.

## Dynamics of MN generation versus off-target differentiation in DP and the SP

Our single cell data allow us to quantify the dynamics of MN generation in DP and the SP over time, by counting the percentage of cells in each state at each time point. Previous reports have claimed up to 50% MN differentiation efficiency for the SP (*Wichterle et al., 2002*; *Wu et al., 2012*), and up to 98% efficiency for DP (after excluding cells that failed to activate the transgene) (*Mazzoni et al., 2013*). However, both of these studies identified MNs as Mnx1+ cells, and for SP this was measured as GFP+ cells in an Mnx1:GFP reporter system. Since GFP is a stable molecule, and Mnx1 is first expressed in early committed MN progenitors, this definition includes cells from early MN progenitors (between PVNP and MNP in our classification) through to LMNs. If we apply these criteria to our day 5 SP data, we find a similar SP efficiency of 42.2%, and an efficiency of 66% for the DP protocol (not pre-filtering for transgene activating cells).

Using single cell data, it is possible to construct more refined criteria for cell states during MN differentiation, that resolve between the least mature MNPs, maturing EMNs, and mature LMNs, by focusing not on a single marker gene, but on the entire transcriptional state (*Figure 2B–E*; *Figure 2—source datas 1–2*). As EMNs mature into LMNs they downregulate progenitor markers including Mnx1, and upregulate markers of terminal differentiation. For DP, MN production was observed as early as day 4 (17.8% EMN), and increased over time to 3.4% EMN and 62.7% LMN by day 11 (*Figure 2F*). A minority of off-target neuron subtypes, glia, mesoderm, and endoderm cells were also identified, together accounting for <25% of the total population at day 11. We cannot rule out that these fates may emerge from cells that failed to activate the DP transgenes. In the SP, by contrast, we observed significantly lower efficiency in production of the most mature MN states. The SP population contained 18% EMNs and 1.1% LMNs at day 5, and just 6.5% EMNs and 2.6% LMNs at day 12 (9.1% total) (*Figure 2G*). These figures are lower than those obtained from Mnx1-GFP scoring, because they do not include the earliest committed MN progenitors, which are yet to downregulate progenitor genes such as Mnx1 and Isl1 or to upregulate neuronal genes such as Tubb3 and Map2, and because they distinguish between early and late MNs by their expression of terminal differentiation markers such as Syn1, Mapt, Celf4 or Gabr1. Our single cell classifications therefore capture varying degrees of maturity among differentiating MNs.

An unexpected result from this analysis was that, in the SP protocol, off-target populations increase in fractional abundance to overshadow the intended MN products. Off-target products included oligodendrocytes (7.2%), astrocytes (8.6%), muscle (30.6%), and stroma (8.9%), that together accounted for 55.3% of the population by day 12. The increase in these populations possibly reflects the ability of cell states such as oligodendrocytes and astrocytes to proliferate, which could progressively dilute the initial output of post-mitotic MNs, severely reducing MN generation efficiency. To test this idea and to gain additional EMN and LMN transcriptional data, we profiled an additional 1372 single cell transcriptomes from a replicate SP experiment at day 10 (*Figure 3—figure supplement 1*; Supplementary methods). At this time point, the SP population contained 34.4% EMNs and 14.9% LMNs, which was higher than at either day 5 or day 12. This is consistent with the hypothesis that SP has an initial output of MNs, which may then be diluted by proliferating off-target lineages.

## The DP differentiation trajectory lacks intermediates expressing Olig2 and Nkx6-1

What are the differentiation paths taken by each protocol? In the SP differentiation path, cells transit through seven states (*Figure 2C and E*). These state transitions parallel patterning events in the embryo (*Wichterle et al., 2002*; *Davis-Dusenbery et al., 2014*; *Jessell, 2000*): cells first commit to the neural lineage (NP; Sox1+/Sox2+), then are posteriorized (PNP; Sox3+/Hoxb8+/Hoxd4+), ventralized (PVNP; Nkx6-1+/Olig2+), enter the committed MN progenitor state (MNP; Mnx1+), and then mature through early (EMN; Mnx1+/Tubb3+/Map2+) and late (LMN; Mnx1-/Tubb3+/Map2+/Syn1+) MN states. This is not a surprise as the growth factor cocktail defining this method was designed to reflect the signaling events taking place in the embryo. By contrast, we found that the path produced by DP was condensed relative to the SP path (*Figure 2B and D*), consisting of only four apparent states as opposed to seven. After neural commitment (NP; Sox1+), cells immediately began expressing committed MN markers (EMN; Mnx1+/Tubb3+). This immediate transition was also registered by time course qPCR measurements for a panel of MN genes across bulk populations (*Figure 2—figure supplement 3*).

A major difference between the DP and SP cell states is the seemingly complete absence of typical spinal embryonic intermediates (PNP or PVNP) in DP. These states are normally recognized by expression of the marker genes Olig2, Nkx6-1, Hoxb8, and Hoxd4, but these genes were not detected by scRNA-Seq in DP intermediates. Olig2 is necessary for MN development in embryos (*Takebayashi et al., 2002*), indicating that DP drives differentiation from ESC into MNs through a fundamentally new route.

To more confidently demonstrate that DP does not transit through an Olig2+ state, even transiently, we conducted a high-sensitivity gene expression analysis. In SP, our scRNA-Seq data reveals that Olig2 expression peaks (in the PVNP cells) at a level 5.4-fold higher than Gapdh, a constitutively expressed 'housekeeping' gene. We performed a dense qPCR time course measuring the expression of Olig2 and a panel of MN genes every day during DP across populations of $\sim 10^6$ cells. These measurements showed that committed MN markers were upregulated immediately following early neural progenitor genes in DP, but Olig2 was not detected at any time point during DP (no qPCR signal at $C_T > 40$), showing that the average expression levels are $10^6$-fold or more lower than Gapdh (*Figure 2—figure supplement 3*). Accordingly, on any day, the fraction of cells expressing Olig2 at levels seen in SP was less than one in $10^6$. Given that the total protocol duration was 11 days, this would require that an Olig2 transitional state would last less than 1 s (11 days/$10^6$), far shorter than the lifetime of mRNA molecules in a cell. Therefore, there is no significant Olig2+ transitional state in DP. This conclusion holds even after allowing for the possibility that Olig2 might act through just a single mRNA transcript. From the qPCR data, the fraction of cells expressing Olig2 at one molecule per cell cannot be more than 0.1% (see Supplementary Methods), and the corresponding lifetime of Olig2+ cells would correspondingly be 15 min (11 days/$10^3$), still inconsistent with the lifetime of a single mRNA molecule. These results are consistent with a previous study that reported Olig2 absence during DP (*Mazzoni et al., 2013*).

## The DP and SP trajectories bifurcate after early neural commitment and converge separately to the terminal MN state

Since the DP omits spinal embryonic intermediates characteristic of the SP path, there must be one of two possible trajectories. Either DP must discontinuously transition from an early neural progenitor into a MN, or it must transit through alternate intermediate state(s). To determine which of these possibilities was the case, we employed a data visualization technique called SPRING (*Weinreb et al., 2016*) to directly compare the topology of both paths. While tSNE is a powerful method for identifying discrete cell states, SPRING provides a complementary description emphasizing continuum gene expression topologies. SPRING builds a k-nearest-neighbor graph over cells in high-dimensional gene expression space, and then renders an interactive 2D visualization of the cell graph using a force directed layout. This representation revealed that the DP and SP trajectories overlap during early neural commitment, but that they then bifurcate and take distinct paths that converge independently to the same MN state (*Figure 3A*). The dynamics of gene expression over these trajectories resembled the behavior inferred using tSNE, with DP omitting intermediate progenitor genes following its bifurcation from the SP path (*Figure 3B*).

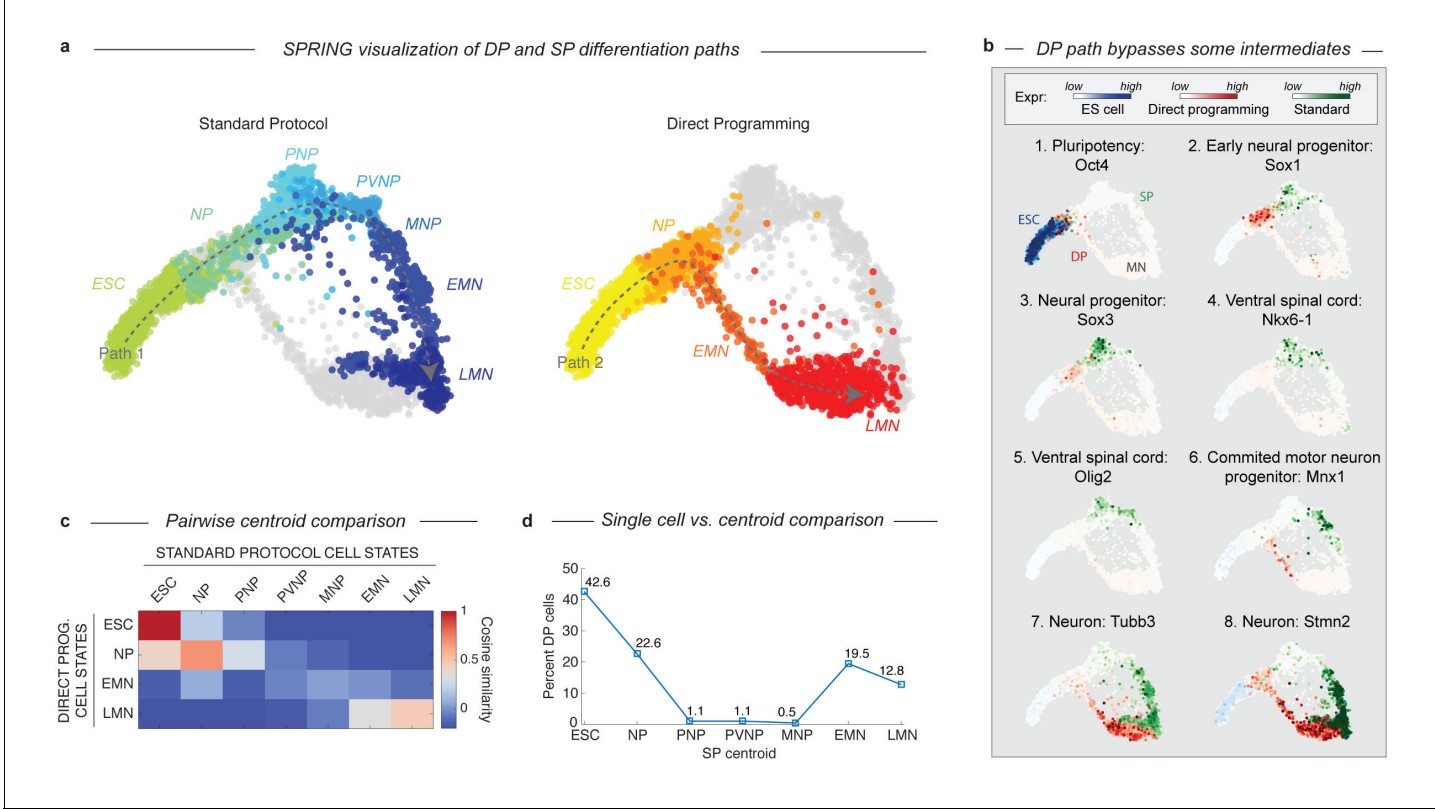

**Figure 3.** DP and SP induce distinct differentiation paths to the motor neuron state. (**a**) Visualization of differentiation paths for both protocols using SPRING reveals two paths to the same state. DP and SP overlap during early neural commitment, but then bifurcate and converge to the same final MN state separately. Reds = direct programming path; Blues = standard protocol path; arrows indicate hypothezed differentiation trajectories. Cell states are colored and labeled by according to their definition in *Figure 1* for comparison. (**b**) Gene expression of key marker genes along each differentiation path confirms exit from the pluripotent state (Oct4) and progression towards the MN state (Tubb3, Stmn2) for both protocols. Only the SP upregulates Olig2 and Nkx6-1, which mark important MN lineage intermediates in the embryo; this occurs following the bifurcation of both paths. Expression in cells from each sample are colored using either a blue (ESC), red (DP), or green (SP) colormap to allow tracking of each path separately. (**c**) Pairwise cosine similarity of cell state centroids. Note early and late similarity of states, but prominent differences during intermediate state transitions. (**d**) Every individual cell of the DP trajectory was assigned to its most similar SP state using a maximum likelihood method. DP cells map to early and late but not intermediate states of the SP cell state progression.

DOI: https://doi.org/10.7554/eLife.26945.009

The following figure supplements are available for figure 3:

**Figure supplement 1.** Consolidation of DP vs SP end-state comparisons with additional SP EMN and LMN cells does not distort joint topology.

DOI: https://doi.org/10.7554/eLife.26945.010

**Figure supplement 2.** Joint visualization of DP and SP differentiation paths by tSNE.

DOI: https://doi.org/10.7554/eLife.26945.011

The bifurcation and subsequent convergence of the two differentiation paths is also revealed by two other complementary analyses. Pairwise cosine similarities between the cell states from both trajectories (*Figure 3C*; Supplementary methods) indicate similarities between the early states (ESC and NP; cosine similarity >0.69) and late states (LMN; cosine similarity = 0.47), but not the intermediate states (PNP, PVNP, and MNP; cosine similarity <−0.15, –0.03, and 0.04 respectively). We also assigned every individual cell along the DP path to its most similar cluster in the SP path using a maximum likelihood method (*Figure 3D*; Supplementary methods). This showed that it was virtually impossible to find a single cell resembling the SP intermediate progenitors in the DP approach. Similarity was again observed only at the early and late states.

## DP transitions through an unexpected intermediate state with forebrain gene expression

The bifurcation of the SP and DP trajectories leads to different intermediate cell states. A total of 26 transcription factors (TFs) are differentially expressed between the DP and SP intermediate states (*Figure 4A*). A majority of these (61%) were involved in an anterior-posterior positional gene expression axis. The SP intermediates were enriched more than 6-fold for nine posterior and spinal TFs including Olig2, Nkx6-1, Lhx3, and six posterior Hox genes with a corrected p-value<0.001. Each of these TFs is expressed in embryonic MNs. By contrast, the DP intermediates were enriched for seven forebrain TFs including Otx2, Otx1, Crx, Six1, Dmrta2, Zic1, and Zic3 at the same stringency, despite the absence of MNs in the forebrain of embryos. Anterior gene expression was previously observed through bulk measurements of DP (*Mazzoni et al., 2013*), and our results reveal that it occurs within a specific subpopulation of cells in the process of differentiating into MNs.

The positional gene expression signature that characterizes the DP intermediate state appears to be transient. Forebrain gene expression is upregulated along the DP differentiation path, as cells exit the early NP state into the EMN intermediate state (*Figure 4B*). This transition is accompanied by the downregulation of proliferation-associated genes (*Figure 4B*; *Figure 4—figure supplement 1*). By the time cells exit the EMN state and transit into the more differentiated LMN state, they downregulate forebrain genes and replace this abnormal positional signature with a spinal Hox expression signature characteristic of normal MNs (*Figure 4B*). Thus cells converge to the MN state in positional as well as neuronal identity gene expression in the final stages of DP.

We validated the existence of the abnormal transitional state in DP in two ways using an Mnx1: GFP reporter cell line. Mnx1 serves as a useful reporter in this context since its expression is localized precisely to the distinct intermediate populations of each differentiation path (*Figure 3B*). First, we performed bulk microarray comparisons of Mnx1+ DP and SP intermediates to compare their gene expression states by an independent method. The comparison confirmed the enrichment of forebrain TFs, and depletion of spinal progenitor and positional genes in the DP intermediates with just one exception – Zic1 was enriched in DP by our single cell comparison but in SP by microarray (*Figure 4A*). Second, we isolated Mnx1+ DP intermediate cells on day 3 of differentiation, the earliest time point at which Mnx1+ cells appear, and prior to maturation of any Mnx1- GFP+ cells. We cultured this population for an additional 10 days, and validated by immunostaining for Vacht and Map2 that it indeed generates motor neurons (*Figure 4C*). This experiment confirms that the abnormal transitional state in DP gives rise to MNs. The abnormal DP intermediate thus simultaneously has a forebrain positional expression state and possesses MN fate potential.

## Both DP and the SP approach a transcriptional state similar to *bona-fide* MNs in embryos

Given that the two protocols induce distinct – and in the case of DP, unnatural – differentiation paths, we were curious how their final products compared with primary MNs (pMNs). We harvested MNs from the embryo of a Mnx1:GFP reporter mouse and performed inDrops measurements on 874 Mnx1+ cells that were FACS purified from whole E13.5 spinal cords. Though the majority of Mnx1+ sorted cells were MNs (73.8%, n = 645), this population also contained glia (20.1%), fibroblast-like cells (1.8%), and immune-type cells (1.2%; *Figure 5A*; *Figure 5—figure supplement 1*). Using only the cells identified as *bona-fide* MNs, we compared the differentiating DP and SP cells to pMNs by both global transcriptome similarity of cell states centroids, and a nearest neighbor analysis of single cells. Global transcriptome comparisons confirmed that each state along the DP and SP differentiation paths becomes progressively more similar to pMNs (*Figure 5B*). The clusters most similar to pMNs were the LMN state from the DP protocol (cosine similarity = 0.60), and the LMN state from the SP (cosine similarity = 0.47). Since subsets of LMNs from DP and the SP might vary in similarity to pMNs, we analyzed the similarity of single cells from all three experiments using SPRING, by embedding all three data sets onto a single kNN graph. We performed this analysis including all cells (*Figure 5C–i*), and then including only EMNs, LMNs, and pMNs (*Figure 5C–ii*). Both approaches showed that pMNs closely associate with the LMNs of both DP and SP. It was also apparent that DP and SP LMNs are themselves heterogeneous, with particular subsets associating more closely with pMNs. Overall, a higher fraction of DP LMNs resembled primary MNs, as seen by calculating the fraction of cells in each state that had at least one pMN nearest neighbor out of its

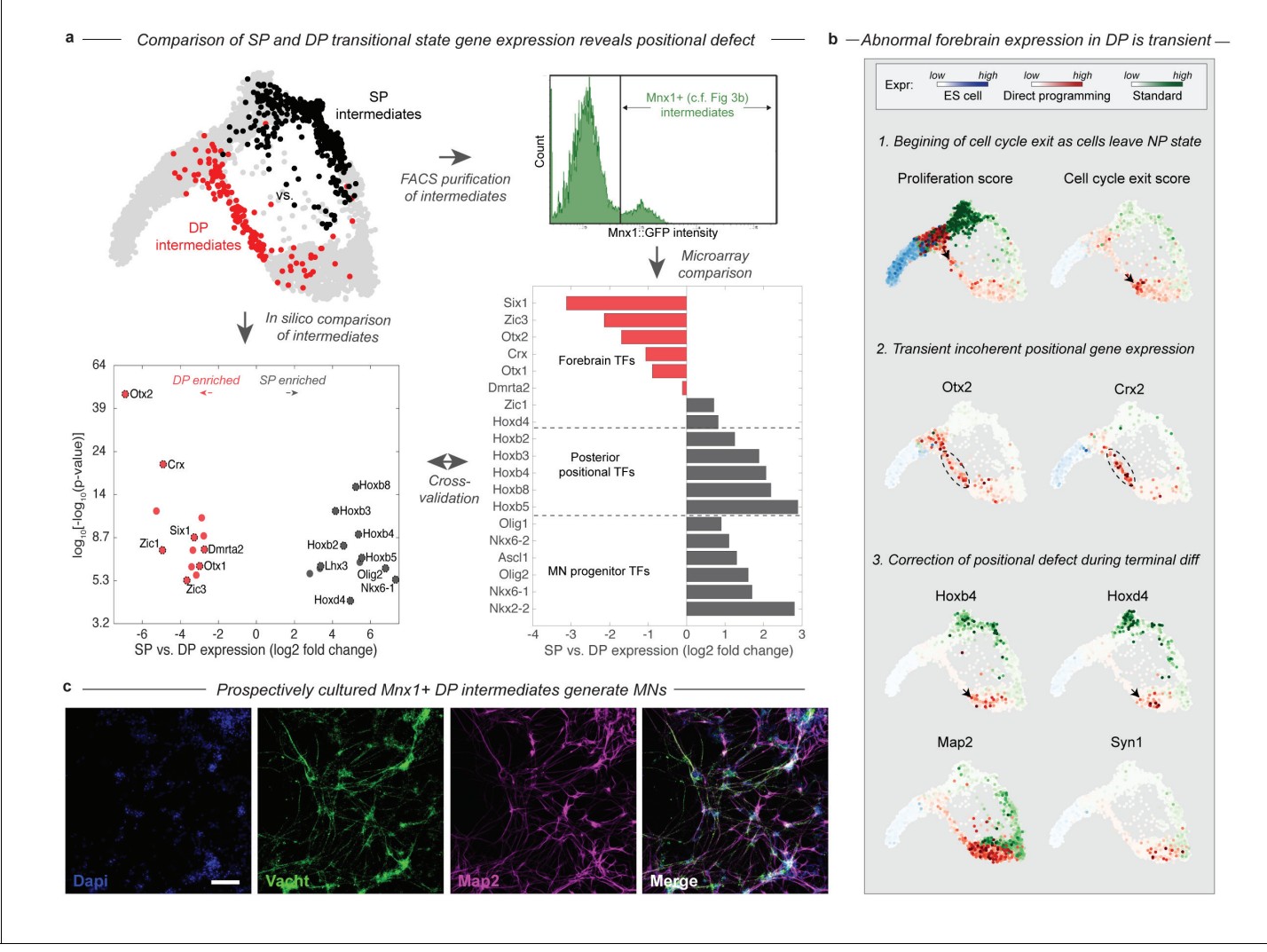

**Figure 4.** During DP cells transition through an abnormal intermediate state with a forebrain gene expression defect. (a) The DP (red, EMN from *Figure 2*) and SP (black, PVNP + MNP + EMN from *Figure 2*) differentiation paths diverge into distinct intermediate states following early neural commitment. These distinct populations were compared by in silico differential expression analysis using single cell data (bottom left), or by FACS purification of the intermediate populations using and Mnx1::GFP reporter cell line followed by microarray analysis (bottom right). *Figure 3b.* shows that Mnx1 expression localizes to the distinct intermediate populations of both protocols, making it an appropriate target for their purification. Differential expression analysis revealed a total of 26 differentially expressed TFs between the intermediate populations (>6 fold differentially expressed, p<0.001), of which (61%) were involved in either forebrain positional identity (Otx2, Crx, Zic1, Six1, Dmrta2, Otx1, and Zic3) and enriched in DP, or in posterior spinal identity (Hoxb8, Hoxb3, Hoxb2, Hoxb4, Hoxb5, Hoxd4, Lhx3, Olig2, and Nkx6-1) and enriched in SP. The independent microarray comparison confirmed these differences with only one exception - Zic1. It also revealed that the SP intermediates are enriched for additional spinal neural progenitor genes including Ascl1, Nkx6-2, Olig1, and Nkx2-2. (b) Before forebrain genes are expressed, DP cells have a high proliferation score and low cell-cycle exit score (computed as the sum of a panel of cell-cycle-associated or tumor supressor genes respectively). They reduce cell cycle gene expression as they enter the abnormal transitional state, and upregulate forebrain genes including Otx2 and Crx. This abnormal forebrain expression is shut off as cells exit the transitional state into the final MN state, characterized by Map2 and Vacht expression. The final transition into a MN state is also accompanied by upregulation of posterior positional genes including Hoxb4 and Hoxd3, thus correcting the transient forebrain positional expression defect. Expression in cells from each sample are colored using either a blue (ESC), red (DP), or green (SP) colormap to allow tracking of each path separately. (c) Mnx1+ cells were isolated at day 3 of differentiation and cultured to determine their fate potential. By 12 days after replating most cells expressed motor neuron markers Map2 and Vacht. Scale bar = 100 um.

DOI: https://doi.org/10.7554/eLife.26945.012

The following figure supplement is available for figure 4:

**Figure supplement 1.** Cell-cycle-associated gene expression decreases as cells enter the DP and SP intermediate states and then progressively mature.

DOI: https://doi.org/10.7554/eLife.26945.013

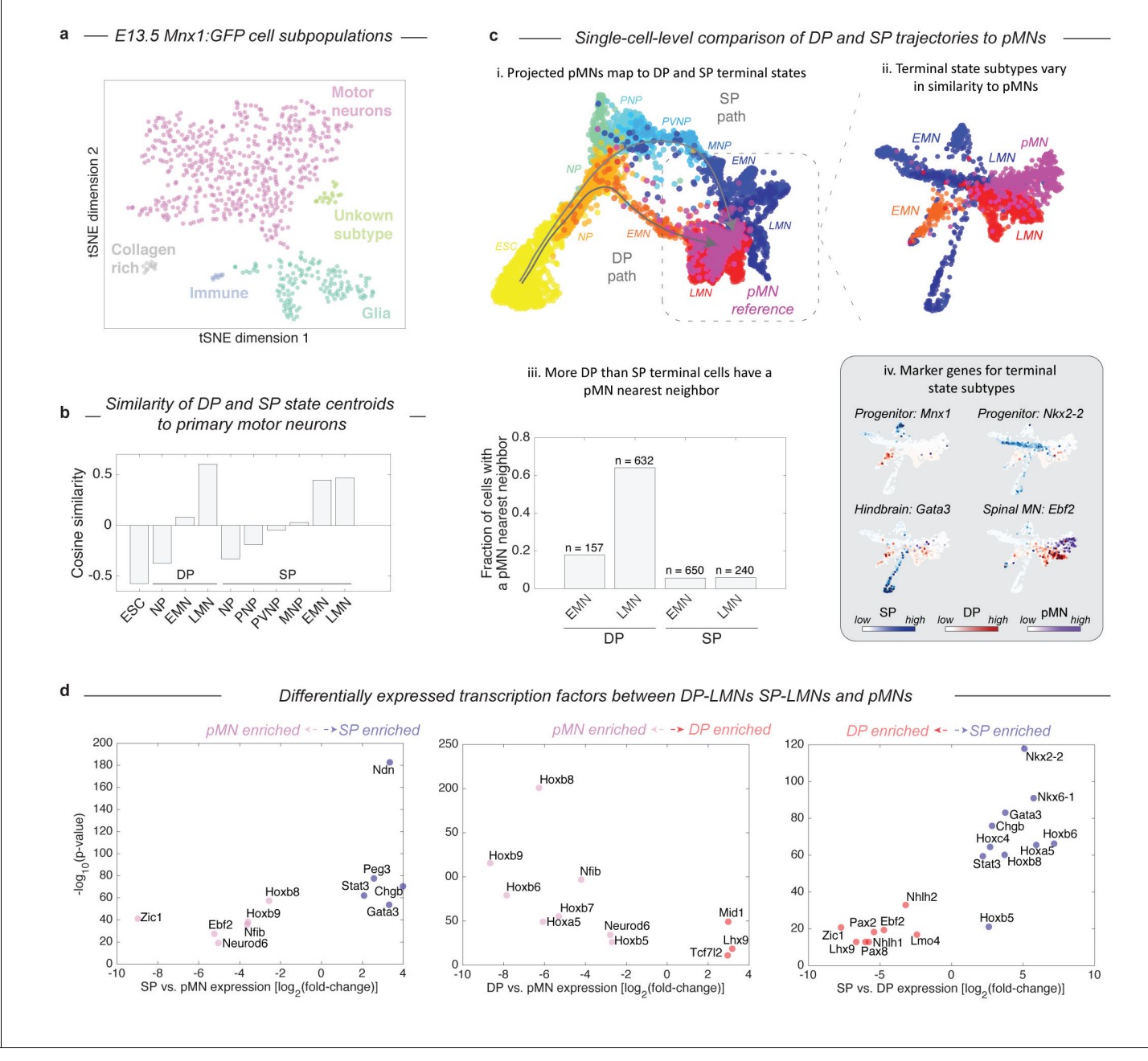

**Figure 5.** Both DP and SP differentiation trajectories approach the transcriptional state of primary MNs (pMNs), but DP does so with higher precision. (a) tSNE visualization of 874 single cell transciptomes from FACS purified Mnx1+ MNs from embryos reveals heterogeneity within this population. To make comparisons between DP and SP with pMNs we used only the subset of Mnx1:GFP+ primary cells in a bona-fide MN state. See *Figure 5—figure supplement 1* for marker gene expression in each population. (b) Comparison of average gene expression profiles for cell states along the DP and SP trajectories with pMNs. In both methods similarity increases as differentiation proceeds. Late DP states are the most similar to embryonic MNs. (c) Projection of the reference E13.5 pMNs into the visualization from *Figure 3* revealed that pMNs closely associate with the terminal states of both DP and SP (i). Close examination of the terminal populations (EMN, LMN) from DP and SP compared to pMNs reveals heterogeneity representing state subtypes (ii). At a single cell level DP LMNs were the most closely associated with E13.5 pMNs; 64% of DP LMNs had at least 1 pMN nearest neighbor out of its most similar 50 cells compared to 6% for SP LMNs (iii). The subtypes present within terminal DP and SP populations could be annotated using marker genes. DP and SP EMNs express progenitor genes including Mnx1, along with Nkx2-2 and Nkx6-1 in SP only. The major SP LMN outgroup expressed Gata3, indicating a hindbrain identity. Both DP LMNs and pMNs shared expression of the terminal MN differentiation gene Ebf2. (d) Systematic pairwise differential gene expression analysis between terminal DP and SP states and pMNs. Each panel is a volcano plot of differentially expressed transcription factors. Both DP and SP LMNs show limited gene expression differences to pMNs. The dominant differences are positional, with DP and SP LMNs lacking the most posterior Hox genes. Other expression differences are explained by differences in terminal state subtypes as

*Figure 5 continued on next page*

*Figure 5 continued*

shown in c). Differentially expressed genes were filtered for TFs with a corrected p-value<0.001, expression ratio > 4, and minimum expression of 1umi/cell average.

DOI: https://doi.org/10.7554/eLife.26945.014

The following figure supplement is available for figure 5:

**Figure supplement 1.** Heterogeneity within the Mnx1+ E13.5 primary motor neuron population.

DOI: https://doi.org/10.7554/eLife.26945.015

50 most similar cells (*Figure 5C–iii*; 64% for DP, 6% for SP). DP LMNs therefore appear if anything more related to pMNs in gene expression than SP LMNs, despite their unusual developmental path.

The differences in similarity of DP and SP derived MNs to primary MNs can be understood in terms of differences in maturity and identity of the cells. As expected, both DP and SP EMNs were less mature than pMNs, as seen by expression of the MN progenitor marker Mnx1 (*Figure 5C–iv*). Among the SP LMNs we detected multiple subpopulations. One subset appeared more anterior than the spinal E13.5 pMNs, as indicated by expression of the hindbrain marker Gata3; the second appeared less mature than pMNs as indicated by residual expression of progenitor genes such as Mnx1 or Nkx2-2; and the final population was closely associated with pMNs (*Figure 5C–iv*). Most DP LMNs by contrast were closely related to pMNs. They lacked the residual progenitor gene expression seen in SP, and they shared with pMNs the expression of genes important to terminal MN differentiation (e.g. Ebf2 and Ebf3). Beyond individual marker genes, we also systematically compared DP LMNs, SP LMNs, and pMNs by pairwise differential gene expression analysis (*Figure 5D*). Overall, both DP and SP LMNs show limited gene expression differences to pMNs. In both cases there were 11 differentially expressed TFs. Of these, half were positional Hox genes that mark the most posterior MN pool and were enriched in pMNs, indicating a more anterior identity of the DP and SP LMNs. The remaining TFs included Gata3, which marks a hindbrain subpopulation of SP LMNs as noted above, and other genes related to neuronal maturation including Neurod6, Ebf2, and Mid1. These comparisons confirm the broad similarity of DP LMNs and SP LMNs, to pMNs, but reveal subtle differences between these terminal states that relate to positional identity and state of maturation.

## DP MNs have structural and functional properties of true MNs

Having established that MNs derived via both gene expression trajectories reach roughly the same MN transcriptional state, we wished to validate that their function and structural organization was also independent of their distinct developmental histories. The SP has been characterized extensively as giving rise to functional MNs (*Wichterle et al., 2002*), so here we examined structural and functional characteristics following DP. We confirmed that selected protein content matches the mRNA markers by immunostaining for Tubb3, Map2, VACht, Isl1, and Hb9 (*Figure 6A*). Tubb3 and Map2 were present, and VACht was seen at discrete puncta on the axons (suggesting localization to acetylcholine secretory vesicles). TFs Isl1 and Hb9 were localized in the nucleus. Finally, the GFP from the Mnx1:GFP reporter was activated and expressed in the cytoplasm. To test the functional properties of the DP MNs, we performed whole-cell patch clamp recordings. Depolarization induced single or multiple action potentials in current-clamp experiments (*Figure 6B*). Depolarizing voltage steps induced fast inward currents and slow outward currents characteristic of sodium and potassium channels, respectively (*Figure 6C*). Exposure to 500 nM Tetrodotoxin (TTX) blocked the inward current, indicating sodium channel involvement. We then tested whether our DP neurons would respond to neurotransmitters that act on MN (*Figure 6D*). Exposure of the neurons to AMPA, kainate, GABA, and glycine (100 μM each) induced in each case inward currents similar to that seen in primary embryonic MNs. To see if the DP neurons could also form neuromuscular junctions, we co-cultured the neurons with differentiated C2C12 skeletal muscle myotubes and incubated them for 7 days. We observed clustering of acetylcholine receptors on the C2C12 myotubes near contact points with the DP neurons, which can be seen with alpha-bungarotoxin (α-BGT), which binds to acetylcholine receptors (*Figure 6E*). We then observed regular contractions of some C2C12 myotubes that began after several days in co-culture (*Figure 6E*, *Figure 6—video 1*). These contractions could be stopped by the addition of 300 μM Tubocurarine (curare), an antagonist of acetylcholine receptors,

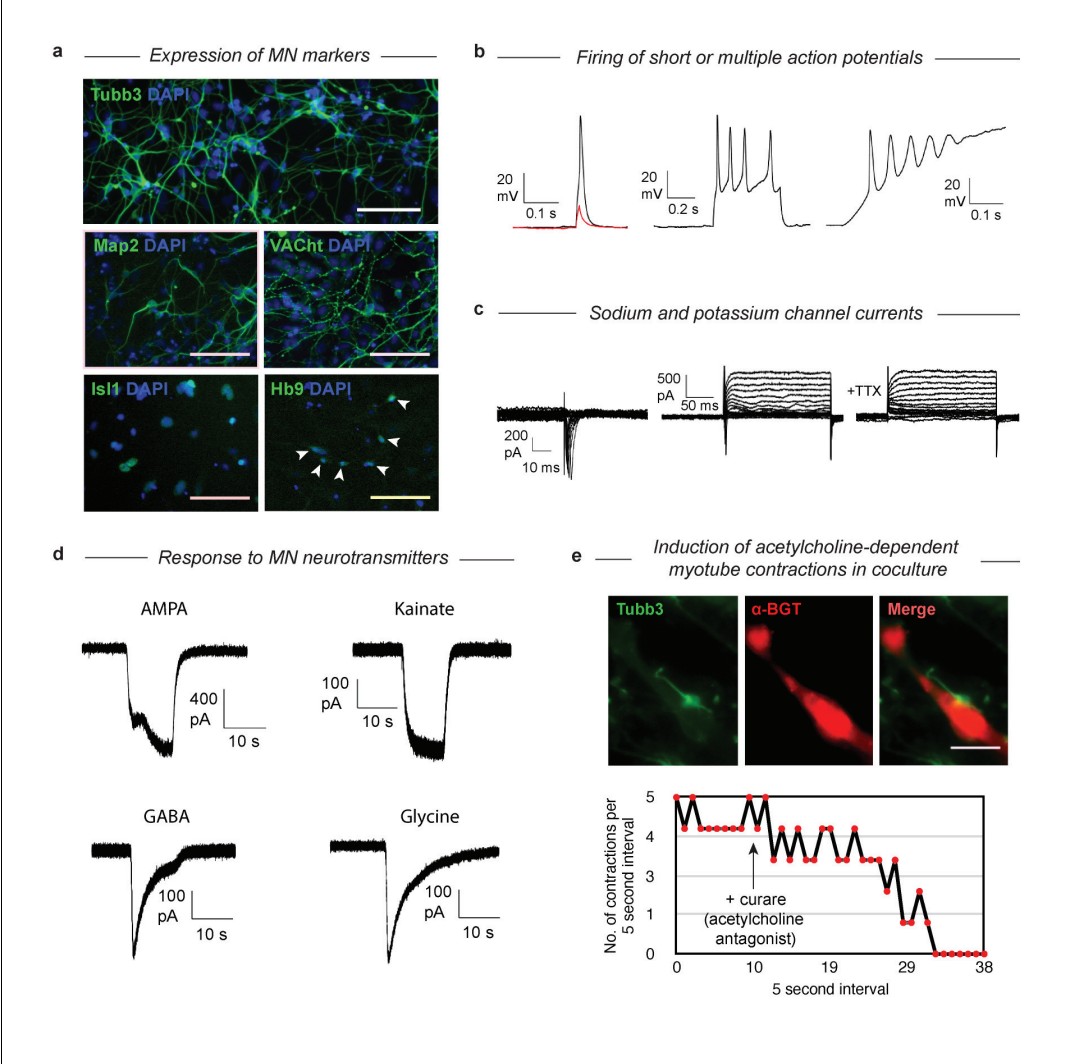

**Figure 6.** Validation that DP cells become functional MNs despite their abnormal differentiation trajectory. (a) Immunostaining of MN markers in DP MNs confirming expression and correct subcellular localization of Tubb3, Map2, VACht, Isl1, and Hb9. DP MNs also: b) can fire single or multiple action potentials upon stimulation, (c) show sodium and potassium channel currents, and d) are responsive to multiple MN neurotransmitters - AMPA, kainate, GABA, and glycine. (e) Co-culture experiments show that DP MNs can induce clustering of acetylcholine receptors on muscle myotubes (indicated by α-BGT staining) and induce their contractions. The induced contractions are dependent on MN activity as they can be blocked by addition of the acetylcholine antagonist curare.

DOI: https://doi.org/10.7554/eLife.26945.016

The following videos are available for figure 6:

**Figure 6—video 1.** Contractions induced by DP MNs in C2C12 myotubes.

DOI: https://doi.org/10.7554/eLife.26945.017

**Figure 6—video 2.** Contractions induced by DP MNs in ES-MyoD myotubes.

DOI: https://doi.org/10.7554/eLife.26945.018

indicating that the contractions were induced by the acetylcholine release from the MNs. Similarly, we noticed that the DP MNs could induce contractions in DP muscle myotubes that we previously generated with MyoD (*Figure 6—video 2*) (*Li and Kirschner, 2014*). These results confirm that DP MNs have the expected functional properties of *bona-fide* MNs.

## Discussion

We provide evidence at the single cell level that differentiation can proceed by multiple routes and yet converge onto similar transcriptional states. While cells differentiated by growth factor stimulation in the SP retrace the embryonic lineage, cells differentiated by terminal state transcription factors in DP take a dramatically different path. The DP path bypasses multiple intermediate progenitor-states that are produced in the embryo, and yet they still converge to the same discrete and recognizable MN phenotype. This convergence occurs via an abnormal intermediate state, and does not appear to involve a shared set of terminal cell state transitions. Moreover, as cells converge they manage to not only establish gene expression related to MN functions, but they also correct positional gene expression defects (exchanging forebrain for spinal gene expression) in the absence of external signals. We conclude that DP of mESCs into MNs occurs via a late bypass that involves alternative intermediate states not seen in the embryo, and that this new route converges near perfectly to the same final state (*Figure 1*). Convergence into a MN therefore does not depend rigidly on the precise history of intermediate states through which cells differentiate.

This 'history independence' of the final state is consistent with a dynamical view of gene regulation in which cell states correspond to 'attractor basins', i.e. stable states of gene expression that are robust to modest perturbations. If attractor basins do not exist, the precision of the observed overlap between DP and SP MNs would require a special coincidence. The concept of cell states behaving as attractors has been proposed previously to explain several properties of blood cell types (*Chang et al., 2008*; *Huang et al., 2007*; *Huang et al., 2005*). There are at least two important corollaries of this behavior applied to motor neuron development. From a practical perspective, it is a common concern that DP methods may generate cell types with subtle defects due to their unusual developmental histories (*Cohen and Melton, 2011*). Attractors would be robust to this vulnerability and indeed our results show that it is not necessary to recreate the precise sequences of steps taken in embryos to generate MNs with the highest similarity to primary cells. It could also hint at a mechanism that might help animal body plans evolve flexibly. Specifically, by decoupling the identity of mature cell state attractors from their developmental histories evolution would be able to act on each independently. In principle this could contribute to evolvability by allowing mature cell states to be transposed onto new lineages in new body locations, without massively rewiring the internal differentiation circuit.

The mechanisms that define the MN attractor basin and allow the artificial DP trajectory to converge onto the correct final state are unclear. The MN state is thought to be stabilized by a network of self-reinforcing TFs (*Jessell, 2000*), involving Mnr2, Mnx1, Lim3, Isl1, Isl2, and Lhx3, Ngn2, Myt1l, Nefl, and Nefm. DP aims to kick-start this network by activating a subset of important components. Yet, far from immediately activating this network, our data show that DP initially drives cells to differentiate into an early NP state through the same pathway as the SP trajectory, seemingly oblivious to the DP TFs, and then even activates non-MN genes in the transitional state. Understanding why the activation of the MN program lags behind TF induction may provide important clues into how the DP factors act. One possible source for a lag is that activating a complete neuronal program requires first activating additional core TFs (so-called 'feed-forward' circuitry). Indeed, recent studies have shown that Ebf and Onecut are activated by Ngn2 (one of the DP TFs), and that both are required to subsequently direct binding of Isl1 and Lhx3 (the other two DP TFs) to MN target genes across the genome during DP (*Velasco et al., 2017*; *Rhee et al., 2016*). A second possible source of lag is that extracellular signaling provides inputs that immediately affect cell state, but take time to sensitize cells to the DP factors. For example, signaling changes might activate DP TFs through post-translational modifications, by activating co-factors, or by inducing chromatin state changes. We have indeed observed that MNs are not generated if DP TFs are induced in cells cultured in pluripotency media, indicating a requirement for changes in signaling (not shown). Conversely, when mES cells are transferred to minimal media without inducing DP factors, they acquire a forebrain neural progenitor identity by default (*Pankratz et al., 2007*; *Smukler et al., 2006*; *Okada et al., 2004*). This suggests that the early dynamics and abnormal forebrain/MN expression of the DP transitional state might in fact be driven by the signaling environment and not the DP TFs. These alternatives suggest future experiments to better resolve the mechanisms driving the DP, by re-mapping the trajectory induced during DP after changing signaling conditions, or the choice of DP TFs.

As a methodology, DP is significantly more efficient than the SP without loss of quality in the MN populations produced (*Figure 2F–G*; *Figure 5*; *Figure 6*). The high-efficiency of DP most likely derives from both its more uniform experimental conditions as well as its more direct differentiation path. Experimentally, DP: relies on 2D rather than 3D tissue culture (as in the SP), minimizing uncontrolled cell-cell communication; forces every individual cell to express MN TFs from a genetically integrated construct, increasing uniformity; and employs a defined-media without growth factors that may minimize proliferation of competing progenitor states. The more direct differentiation path induced by DP might also itself increase MN conversion efficiency by minimizing error propagation through off-target fate choices. Long sequences of intermediate cell state transitions offer many sequential opportunities for off-target fate choices, which multiplicatively reduce efficiency, as compared to the more compact DP trajectory. Targeting terminal attractor basins through the shortest possible differentiation paths may prove to be a generally effective strategy for generating desired cell states, more quickly and more enriched in the desired product.

## Materials and methods

### ESC culture
The mouse ES cell line containing doxycycline-inducible Ngn2+Isl1+Lhx3 (NIL) and the Hb9::GFP reporter was graciously provided by Esteban Mazzoni. ESCs were cultured in standard media (DMEM with LIF + 15% fetal bovine serum) on 0.2% gelatin-coated dishes.

### Differentiation into motor neurons: direct and standard programming
Twenty-four hours before starting differentiation, ESCs were trypsinized and seeded onto plates pre-coated with a mix of poly-d-lysine (100 µg/ml) and laminin (50 µg/ml) instead of gelatin for adherence. ESCs were counted by a Beckman Coulter Counter and seeded at a density of approximately 200,000 cells per well of a 6-well dish. At day 0 (ESCs), the media was switched from standard ES media to N2B27 media (Invitrogen). Doxycycline was also added at 3 µg/ml starting to induce expression of the NIL transcription factors. Media was changed daily. For the standard programming protocol, the steps described in Wu et al. were followed strictly. The Wu et al. protocol involves a period of neural induction using embryoid body culture before subsequent replating on laminin. The culture media is serum containing base media supplemented with specific cocktails of growth factors that change during the early, middle, and late stages of differentiation (*Figure 2A*).

### Mouse embryonic motor neuron cultures
The B6.Cg-Tg(Hlxb9-GFP)1Tmj/J mice (JAX# 005029) were bred with C57BL/6J (JAX# 000664) for embryonic motor neurons dissection. All animal protocols were approved the Institutional Animal Care and Use Committee at Boston Children's Hospital. On gestational day 13 (E13), the female mice were anesthetized and all embryos were collected by caesarian section. Only GFP embryos were used for further dissection. The spinal cords were isolated and their meninges were removed. Each isolated spinal cord was dissociated by trypsin and mechanical trituration. After filtering the cells with 100 µm strainers, the cells were spun down and re-suspended in PBS, and subjected to flow cytometry. Cells were run through a 100 µm nozzle at low pressure (20 p.s.i.) on a BD FACSaria II machine (Becton Dickinson, USA). A neural density filter (2.0 setting) was used to allow visualization of large cells.

### Single cell transcriptomics using InDrops
We dissociated differentiating mESC cultures using a 0.25% Trypsin 2 mM EDTA solution (Gibco). Primary HB9+ sorted motor neurons were dissociated as above. Dissociated cell suspensions were verified to be monodisperse and of viability >95% using a coulter counter with trypan blue staining (BioRad). We then performed droplet-based barcoding reverse transcription (RT) reactions and prepared massively multiplexed sequencing libraries using InDrops as described in Klein et al. Briefly: cells, lysis and RT reagents, and barcoding primers attached to hydrogel beads are combined in nanoliter sized droplets suspended in an oil emulsion using the InDrops microfluidic device. A barcoding RT reaction is then carried out in the droplet emulsion, uniquely labeling the RNA contents of every cell using a cell barcode and unique molecular identifier (UMI). Following RT, barcoded

emulsions are split into batches, and the emulsion is broken. Combined material is amplified into a nanomolar Illumina sequencing library through a series of bulk reactions: second strand synthesis, in vitro transcription (IVT), fragmentation, RT2, and a final low cycle number PCR. The majority of the amplification takes place during IVT ensuring uniform library coverage. Single cell libraries were then sequenced on either the Illumina HiSeq or NextSeq platforms. Reads were demultiplexed using an updated version of the custom bioinformatics pipeline described in Klein et al.. A python implementation of this pipeline is now publically available on GitHub. Briefly, it filters for reads with the expected barcode structure, splits reads according to their cell barcode, aligns them to a reference transcriptome (we used GRCm38 with some added mitochondrial genome transcripts), and then counts the number of different UMIs appearing for each gene in each cell. The final output is a counts matrix of cells vs. genes that we loaded into MATLAB for further analysis.

## Single-cell data clean up: minimum expression threshold, total count normalization, stressed cell removal

Before performing the analyses described below, three steps were taken to ensure that the data was of high quality. First, we required all cells to have at least 1000 UMIs detected. This removed any signal potentially coming from empty droplets. Second, data were total count normalized to ensure differences between cells were not due to technical variation in mRNA capture efficiency or cell size. Finally, cells that had a high stress gene signature were excluded from analysis. Stressed cells were initially recognized as a small percentage of cells (<10%) that clustered apart from everything else and specifically expressed very high levels of a mitochondrial gene set that is associated with cellular stress. Masks that convert raw counts into our filtered set are provided online.

## Visualization of single-cell data using tSNE

To visualize high-dimensional single cell data, dimensionality reduction is essential. We chose to implement tSNE as described in Klein et al. The core steps are summarized as follows. Steps 1–2 preceed tSNE, and focus the algorithm on genes that best describe differences between cell populations.

a. Extract the top 1000 highly variable genes. We do this using a statistical test derived specifically for InDrops data (Klein et al.).
b. Extract principal variable genes. Principal variable genes are a subset of the highly variable genes from step two that we find best describes the cell population structure. The steps to find principal variable genes are:
   1. a.Perform PCA using the top 1000 biologically variable genes.
   2. b.Identify the number of non-trivial principal components. This is done by comparing the eigenvalue of each principal component from a. to the eigenvalue distribution for the same data after being randomized. Only principal components with eigenvalues higher than those observed on random data are retained.
   3. c.Extract genes that contribute most highly to these principal components by imposing a threshold on the gene loadings for each non-trivial PC.
c. 3.Perform tSNE. We used the MATLAB implementation of tSNE from Van de Marteen et al. As input, we supplied z-score normalized principal variable genes, and asked tSNE to perform an initial PCA to a number of dimensions equal to the number of non-trivial principal components in step 4. tSNE then takes cells embedded in this PCA space and nonlinearly projects them onto two dimensions for visualization.

## Subpopulation analysis

The first goal of our single cell analysis was to describe the identity and proportions of cell states generated by each of two motor neuron differentiation strategies. For this purpose we found that good results were achieved using local-density gradient clustering on the 2D tSNE representation of the data. This approach provided a clear and natural cell state classification that was well aligned with prior knowledge about marker gene expression domains. The steps we took are summarized as follows:

1. 1.Perform tSNE (as above) on cells pooled from all timepoints for each protocol (e.g. *Figure 2B*).
2. 2.Apply local density gradient clustering to define cell states.

3. 3.Identify genes specifically expressed by each cell state, and use prior knowledge on their expression domains to generate a cell type annotation (e.g. *Figure 2D*).
4. 4.Quantify the fraction of cells in each state at each timepoint (e.g. *Figure 2F*).

We identified genes that were specifically expressed by each subpopulation through pairwise t-tests and visually inspected their expression over the tSNE embedding. In *Figure 2D–E* of the main text, we show the z-score normalized expression of a selection of marker genes that we used as a heatmap. Z-score normalization preserves differences between populations while putting the expression level of every gene on the same scale. In *Figure 2—figure supplements 1–2*, we show the un-normalized expression of each of these genes individually so that the reader may compare.

## Initial identification of differentiation trajectories

During our subpopulation analysis we observed, for both protocols, a continuous progression of cells that spanned the initial ES cell state through the early and late differentiation timepoints. This progression was punctuated by familiar progenitor states that were ordered in a way that was consistent with known events in motor neuron differentiation (*Figure 2D–E*), and was correlated with chronology (*Figure 2B–C* inset). We interpreted these progressions as differentiation trajectories. Each is reconstructed from three population snapshots (day 0, day 4/5, and day 11/12). Because differentiation in vitro is asynchronous, the single cell data overlapped from one timepoint to the next. We deduce from this that intermediate states were not missed due to the spacing of our timepoints. We also validated that important intermediate genes were not detected over a densely sampled time-course using qPCR (see below).

## Differential gene expression analysis between cell states from single-cell data

We identified differentially expressed genes between cell states by using two-tailed t-tests with a multiple hypothesis testing correction. We defined differentially expressed genes conservatively at a FDR of 5% and a significance level of $p<0.0001$. We only considered genes where at least one of the states being compared had >= 10 cells with non-zero expression. To find marker genes of a population we asked for genes that were enriched in that population versus everything else. For several comparisons we restricted our analysis to Riken transcription factors. This list contains ~1500 genes with manually annotated transcription factor activity. We represented differential expression data using volcano plots, and colored the expression intensity of each gene using a colorbar; the mean of the higher expressing state was used.

## Integration of replicate SP EMN/LMN cells into original SP trajectory

Due to the low efficiency of EMN/LMN production in our initial SP experiment, we performed a replicate experiment with the goal of extracting additional SP terminal state cells. We repeated SP differentiation and performed InDrops on an additional 1372 cells sample at day 10 of differentiation. To identify EMN or LMN cells within this population we: projected replicate SP cells into the PC-space from the original experiment; calculated a Euclidean distance matrix between all cells; extracted replicate SP cells which had >= 3 of their five nearest neighbors belonging to EMN or LMN clusters; then finally assigned cluster identities to these cells such that cells were called EMN if a majority of their neighbors were EMN, or otherwise were called LMN (majority LMN nearest neighbors or equal EMN and LMN nearest neighbors). This final rule slightly favors LMN over EMN cluster assignment but this helps to counter the low LMN efficiency of the original experiment, which reduces the possible nearest neighbors available during label assignment. For all comparisons of DP and SP we pooled EMN and LMN cells across the two replicate experiments for SP, improving the statistics of the end state comparisons.

## Comparison of differentiation paths I: Visualization of alternate differentiation trajectories using SPRING

After our initial identification of the differentiation paths for the standard protocol and for direct programming we wished compare their routes. One of the most powerful ways to begin addressing such a problem is to simply visualize the data. Yet, we found tSNE gave unclear results for a direct comparison of the paths; visualizing both protocols together resulted in some mixed clusters, and

some distinct clusters, but no overall coherence to the representation as we had found looking at one or the other trajectory separately. The limitations of tSNE for analyzing continuous processes are well known.

We therefore turned to a new method developed in parallel to this work in our lab called SPRING (*Weinreb et al., 2016*), that in our experience does better in analyzing continuous processes in single cell data. SPRING has four core steps: first, it filters for genes with average expression >0.02 UMIs/cell, and Fano factor >2; second, it reduces dimensionality to a 50 dim PCA space; third, it constructs a k-nearest-neighbor (kNN) graph in this space; finally, it renders an interactive 2D visualization of this kNN graph using a force directed layout. In this visualization edges of the kNN graph are literally springs that pull together similar cells, while every cell has a magnetic repulsion force that pushes it away from other cells and an intrinsic gravity. The balance of these forces illuminates the topology of how cells are positioned in high-dimension with respect to one another. In this visualization nodes can be moved around to rotate the projection and find the clearest representation; in each new position the graph re-relaxes according to the underlying physics of the force directed layout. The specific steps we took to generate *Figure 3A* are as follows. Note that a small manual correction was made to the position of the ESC and LMN populations to reduce white space, but did not distort relationships between populations.

1. 1.Load the cells along each differentiation path into SPRING separately.
2. Remove doublets. Doublets were identified by three criteria. (1) they are rare, in line with our experimental expectation for cell doublets, (2) they formed long range connections between large cell groups in the same timepoint and sample, (3) they do not possess any unique marker genes; all genes they express are a linear combination of two other cell states. We identified approximately 80 doublets in the standard protocol (<3% of all cells), and approximately 40 in the direct programming approach (<2% of all cells).
3. Load the filtered cells from both protocols into SPRING together. To make plots the coordinates from the SPRING representation were exported into MATLAB; cells were either colored by cell state (*Figure 3A*), or gene expression (*Figure 3B*).

## Comparison of differentiation paths II: Pairwise cosine similarity of cell state centroids

We also asked how the direct programming and standard motor neuron differentiation paths were related to each other by performing a pairwise comparison of cell state centroids. Centroids are the average or center of mass in gene expression space for a collection of cells. Centroids can provide a very accurate estimate of global gene expression that averages over the noise intrinsic to single-cell RNA sequencing at the level of individual cells. By computing the cosine similarity between two cell state centroids we are asking how similar these states are in average gene expression. If the paths do indeed split and then reconverge our expectation was to see similarity between early and late states in both progressions, but not between the intermediate states. We chose to perform these comparisons in a PV-gene space constructed from cells in both protocols to make our comparisons as sensitive as possible. We obtained similar but less sensitive results using all genes above a minimum expression value (not shown). A summary of the steps of this analysis is:

1. Combine all cells from both protocols, extract PV-genes (as described above), and z-score normalize their expression.
2. Calculate the centroid of every cluster from each trajectory in this PV-gene space.
3. Compute the pairwise cosine similarity for all direct programming clusters versus all standard protocol clusters.
4. Visualize: we chose to use a heatmap (*Figure 3C*).

## Comparison of differentiation paths III: Maximum likelihood assignment of single cells to cell state centroids between trajectories

Finally, we performed an independent comparison of the differentiation paths that did not depend on our definition of cluster boundaries in the direct programming trajectory. We asked whether any potentially rare individual cells or subpopulations in the direct programming trajectory resemble the intermediate states of the standard trajectory. Because we were now dealing with single cells, not averages of many measurements, care was necessary in this analysis to remain robust relative to the

noise in single cell measurements. We therefore took a Bayesian approach and reasoned as follows. In our data, each cell is a vector of counts, with one element for every gene. These counts can be viewed as a multinomial sample from some underlying distribution of gene expression. Since each state of the standard differentiation trajectory is defined by a particular gene expression distribution, we can ask: what is the probability a given direct programming protocol cell was sampled from each standard differentiation trajectory cluster? In this usage, probability amounts to a measure of similarity, with high probability indicating high similarity. We obtained similar results working in either a PV-gene expression space, or considering all genes expressed above a minimum counts threshold. The results of this analysis using PV-genes are presented in *Figure 3D*. We also obtained similar (but more noisy) results using cosine similarity as an alternative distance metric (not shown). The specific steps of our computation are as follows:

1. Extract a set of genes with which to make comparisons (either PV-genes or all genes above a minimum expression threshold).
2. For each cluster of the standard trajectory, calculate the probability of observing a given gene (i.e. the fraction of counts in that cluster from the gene). For genes that are not detected add 1e-07 total counts.
3. For each cell in the direct programming trajectory, calculate the log-likelihood that it was drawn from each of these clusters. This log-likelihood is from the multinomial distribution function using the probabilities obtained in step 2.
4. 4.Identify and tally the maximum likelihood assignments of all direct programming cells. Normalize raw assignments so that they sum to 100 (giving the percentage). Plot the percentage of direct programming cells assigned to each standard protocol state.

## Cell cycle gene expression analysis

Cell-cycle activity can be estimated from a cell cycle associated gene expression signature; populations that express higher average levels of cell cycle genes are most likely cycling at higher frequency than a population with lower level expression of these genes. In *Figure 4b* and *Figure 4—figure supplement 1* we performed an analysis in this spirit to determine which parts of the DP and SP differentiation trajectories appeared to be proliferative, and to estimate where cells are exiting the cell cycle. We computed a proliferation score that was the aggregate expression of a panel of 21 cell cycle related genes: Aurka, Top2a, Ccna2, Ccnd1, Ccnd2, Ccnd3, Ccne1, Ccne2, Ccnb1, Cdk4, Cdk6, Cdk2, Cdk1, Cdkn2b, Cdkn2a, Cdkn2c, Cdkn2d, Cdkn1a, Cdkn1b, Cdkn1c, Mcm6, Cdc20, Plk1, and Pcna. We also computed a cell cycle exit score on the basis of the aggregate expression of a panel of 4 tumor suppressor genes that inhibit the cell cycle: Cdkn1c, Cdkn1b, Cdkn1a, and Cdkn2d. In *Figure 4—figure supplement 1* we show the expression of representative individual genes from this score; in general cell cycle genes were correlated with each other in their expression over cells, as were cell cycle exit genes.

## Lifetime estimate for hypothetical Olig2+ intermediate cells in DP

We were interested to convert a bound on the population expression of Olig2, obtained by qPCR, into a bound on the lifetime of a hypothetical rare subpopulation that expressed appreciable levels of this transcript. To proceed, we noted that in SP, Olig2 is expressed 5-fold higher than Gapdh. By contrast, qPCR indicated that in DP, Olig2 was expressed $10^6$-fold lower than Gapdh. Since we took qPCR measurements on every day during differentiation, we next reasoned that the maximum possible number of these intermediate cells would exist if they were spread uniformly across our timecourse (otherwise there would be a spike in their number, increasing the chances of their detection). This is unrealistic, but sets a conservative upper bound. Since the total differentiation protocol took 11 days, any Olig2+ intermediate that expressed the gene at the levels seen in SP (five fold higher than Gapdh) must thus exist for less than 11 days/$10^6$/5, or less than 0.2 s. To estimate what this lifetime would be for an Olig2+ population that expressed just one molecule per cell, we conservatively allowed for Gapdh expression levels of 1000 molecules per cell (smRNA-FISH studies have estimated ~200–500 Gapdh copies per cell). The maximum fraction of Olig2+ cells expressing one molecule per cell at any timepoint in DP is correspondingly 0.1% (1000/$10^6$*100%). Our lifetime calculation becomes 11 days/$10^3$, or less than 15 min. Since the timescales of mRNA production and

degradation are typically on the order of hours, we therefore concluded that an Olig2+-subpopulation must not exist during DP.

## Comparison of motor neurons in vitro with primary motor neurons using single-cell data

How does the transcriptional state of motor neurons produced by both protocols compare to that of motor neurons in vivo? To answer this question we leveraged the ability of single-cell RNA sequencing to compare cell states even within populations that are not pure (also see below for functional comparisons of the phenotypes). We performed three analyses. First, we computed the cosine similarity between the centroids of each cell state in both protocols and primary motor neurons (*Figure 5B*). The specific steps of this analysis were as follows:

1. 1.Combine all cell states from both protocols, and from HB9+ E13.5 primary tissue, extract PV-genes (as described above), and z-score normalize their expression.
2. 2.Calculate the centroid of every cluster from each trajectory, and from primary motor neurons, in this PV-gene space.
3. Compute the cosine similarity for all in vitro populations versus primary motor neurons, and visualize as a bar graph.

Second, we compared single cells from the DP and SP terminal states to pMN reference single cells. The goal of this analysis was to detect whether heterogeneity within the DP and SP terminal states correlated with similarity to pMNs. To gain a broad understanding of the relationship between the DP and SP trajectories to our pMN reference, we initially projected pMN single cells into the PCA-space used in the visualization from *Figure 3A* (*Figure 5C–i*). The projection was performed by multiplying the pMN counts matrix (after quality filtering cells and z-score normalizing genes) by the PC-loadings matrix that was built on the DP and SP cells as described above. The combined coordinates of all cells in this space were then used as input to SPRING, which builds and visualizes a kNN graph. To generate a more refined visualization of how the terminal populations were related to each other we next constructed a new PC-space on EMN and LMN DP and SP cells, combined with pMNs, following the same steps of SPRING visualization described above (*Figure 5C–ii*). This differs from the original projection in that the second PC-space was constructed to reflect variation within only the mature populations of cells (EMN, LMN and pMN). It thus has higher sensitivity to detect subtle differences between MN subpopulations, because the PC-space does not contain dimensions that describe unrelated process that occur in the process of becoming a neuron from an mES transcriptional state, and thus only introduce noise into the comparison of terminal states. To quantify how similar individual cells were to the pMN reference we performed nearest neighbor analysis on the distance matrix underlying this second SPRING visualization. For each DP or SP terminal state, we asked of every cell whether it had a pMN cell among its 50 most similar cells, and found the fraction of cells with similarity to pMNs. The number of nearest neighbors chosen, in this case 50, sets the size of the region in which we required a pMN cell to fall in order for a cell to be classified as similar to pMN. 50 cells equated to roughly 2% of the total cells in this visualization, and so provided an appropriate and stringent threshold.

Third, we performed differential gene expression analysis, comparing the most mature motor neuron states from each protocol with primary HB9+ motor neurons (*Figure 5D*). This analysis was performed as described above.

## qPCR

RNA was isolated using Qiagen Rneasy Plus Kit. Purified RNA was then reverse transcribed using Bio-rad's iSCRIPT cDNA synthesis kit. Quantitative PCR was performed using Bio-rad's SYBR green Supermix on a CFX96 Real-Time PCR system.

## Immunostaining

Antibodies used for immunostaining were anti-Tubb3 (Cell Signaling D71G9; 1:100), anti-Map2 (Sigma M9942; 1:200), anti-VACht (Synaptic Systems 139 103; 1:200), anti-Isl1 (Abcam ab109517; 1:1000), and anti-Hb9 (DSHB 81.5C10; 1:100). Differentiated cells were fixed in 4% paraformaldehyde for 20 min and then permeabilized with 0.1% Triton-X for 15 min. After primary incubation for

1 hr, samples were labeled with a secondary antibody conjugated to AlexaFluor647. Samples were co-stained with DAPI before imaging on a Nikon Eclipse TE2000-E microscope.

## Electrophysiology

All recordings were carried out at room temperature within 6 days of plating the neurons in 35 mm dish. Whole-cell voltage clamp recordings were made with a Multiclamp 700B amplifier (Molecular Devices) and patch pipettes with resistances of 2–3 M$\Omega$. Pipette solution was 135 mM K-Gluconate, 10 mM KCl, 1 mM $MgCl_2$, 5 mM EGTA, 10 mM HEPES, pH 7.2, adjusted with NaOH. The external solution was 140 mM NaCl, 5 mM KCl, 2 mM $CaCl_2$, 2 mM $MgCl_2$, 10 mM HEPES, and 10 mM D-glucose, pH 7.4, adjusted with NaOH. We used gravity perfusion system connected with Perfusion Pencil with Multi-Barrel Manifold Tip (AutoMate Scientific) to externally apply 0.5 µM tetrodotoxin, 100 µM AMPA, kainate, GABA, or glycine to the cells. Command protocols were generated and data was digitized with a Digidata 1440A A/D interface with pCLAMP10 software.

## Co-culture muscle contraction assays

C2C12 myoblasts were grown in 10%FBS + DMEM media and then differentiated into myotubes by incubating in differentiation medium (2% horse serum + DMEM). After myotubes were formed, the neurons were dissociated by trypsinization and reseeded on top of the differentiated muscle to allow contractions to develop. Video of contractions were taken using Metamorph software and manually counted over 5 s intervals. For stopping assay, 300 µM Tubocurarine (Sigma) was added to the media as an acetylcholine competitor. For labeling of acetylcholine receptors, bungarotoxin (Invitrogen) was used after cells were fixed with paraformaldehyde. Similarly to C2C12 cells, ES cells over-expressing MyoD were also differentiated to myotubes using differentiation medium and subjected to co-culture with neurons.

## Microarray analysis

The mRNA of undifferentiated iNIL ES cells and NIH 3T3 cells (grow in DMEM + 10% FBS) were collected and purified by RNA extraction using RNeasy Plus Extraction Kit (Qiagen). Neurons differentiated by either protocol were first sorted by flow cytometry on a BD FACSaria II machine (Beckton Dickinson, USA) to collect Hb9::GFP+ cells, and then were subjected to RNA extraction in a similar fashion. Collected RNA was then amplified and hybridized to Affymetrix GeneChip Mouse Transcriptome Arrays (MTA 1.0). Results were processed by the Children's Hospital Microarray Core Facility, and were analyzed using Affrymetrix's Transcriptome Analysis Console and Expression Console software.

## Acknowledgements

We are grateful to Esteban Mazzoni for providing the transcription factor cassette containing ES cell line used in the DP experiments presented here.

## Additional information

### Competing interests

Victor C Li: is co founder of StemCellerant, LLC. Allon Klein: is co founder of 1CellBio, Inc. Marc W Kirschner: is co founder of StemCellerant, LLC and 1CellBio, Inc. The other authors declare that no competing interests exist.

### Funding

| Funder | Grant reference number | Author |
| --- | --- | --- |
| National Institutes of Health | R21 HD087723 | Victor C Li<br>Marc W Kirschner |
| Harvard Brain Initiative | HBI ALS Seed Grant | Allon Klein |

The funders had no role in study design, data collection and interpretation, or the decision to submit the work for publication.

### Author contributions
James Alexander Briggs, Conceptualization, Data curation, Formal analysis, Validation, Investigation, Visualization, Methodology, Writing—original draft, Writing—review and editing; Victor C Li, Conceptualization, Data curation, Validation, Investigation, Methodology, Writing—original draft, Writing—review and editing; Seungkyu Lee, Validation, Methodology; Clifford J Woolf, Supervision, Writing—review and editing; Allon Klein, Conceptualization, Resources, Formal analysis, Supervision, Funding acquisition, Investigation, Project administration, Writing—review and editing; Marc W Kirschner, Conceptualization, Resources, Supervision, Funding acquisition, Investigation, Project administration, Writing—review and editing

### Author ORCIDs
James Alexander Briggs [ID] http://orcid.org/0000-0001-6849-4156
Marc W Kirschner [ID] http://orcid.org/0000-0001-6540-6130

### Ethics
Animal experimentation: This study was performed in strict accordance with the recommendations in the Guide for the Care and Use of Laboratory Animals of the National Institutes of Health. All of the animals were handled according to approved institutional animal care and use committee (IACUC) protocols (#16-01-3080R) of Boston Children's Hospital, and (#IS00000137, #1648, #1648a, #1648b) of Harvard Medical School. The Hb9-GFP embryos were collected from a pregnant mouse after euthanasia by inhalation of $CO_2$, and every effort was made to minimize suffering.

### Decision letter and Author response
Decision letter https://doi.org/10.7554/eLife.26945.027
Author response https://doi.org/10.7554/eLife.26945.028

## Additional files

### Supplementary files
• Transparent reporting form
DOI: https://doi.org/10.7554/eLife.26945.019

### Major datasets
The following datasets were generated:

| Author(s) | Year | Dataset title | Dataset URL | Database, license, and accessibility information |
|---|---|---|---|---|
| Briggs JA, Li VC, Lee S, Woolf, CJ, Klein AM, Kirschner MW | 2017 | Single cell RNA-seq data for: Mouse embryonic stem cells can differentiate via multiple paths to the same state | https://www.ncbi.nlm.nih.gov/geo/query/acc.cgi?acc=GSE97391 | Publicly available at NCBI Gene Expression Omnibus (accession no: GSE97391) |
| Briggs JA, Li VC, Lee S, Woolf, CJ, Klein AM, Kirschner MW | 2017 | Microarray data for: Mouse embryonic stem cells can differentiate via multiple paths to the same state | https://www.ncbi.nlm.nih.gov/geo/query/acc.cgi?acc=GSE97391 | Publicly available at NCBI Gene Expression Omnibus (accession no: GSE97391) |

The following previously published dataset was used:

| Author(s) | Year | Dataset title | Dataset URL | Database, license, and accessibility information |
|---|---|---|---|---|
| Klein AM, Mazutis L, Akartuna I, Tallapragada N, Veres A, Li V, Peshkin L, Weitz DA, Kirschner MW | 2015 | Droplet barcoding for single cell transcriptomics applied to embryonic stem cells | https://www.ncbi.nlm. nih.gov/geo/query/acc. cgi?acc=GSE65525 | Publicly available at the NCBI Gene Expression Omnibus (accession no: GSE65525) |

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
