## [Decision Letter]

Thank you for submitting your article "Mouse embryonic stem cells can differentiate via multiple paths to the same state" for consideration by *eLife*. Your article has been reviewed by three peer reviewers, one of whom is a member of our Board of Reviewing Editors and the evaluation has been overseen by Marianne Bronner as the Senior Editor. The reviewers have opted to remain anonymous.

The reviewers have discussed the reviews with one another and the Reviewing Editor has drafted this decision to help you prepare a revised submission. Essential revisions are listed below and we have also included the detailed reviewer comments for your information.

Summary:

This manuscript uses single cell transcriptomics to compare the cell state transitions that occur when mouse embryonic stem cells are directly reprogrammed into motor neurons using defined transcription factors (direct protocol) to those that occur during a directed differentiation designed to simulate normal embryological induction processes (standard protocol). The main conclusion is that the two pathways converge on a similar endpoint via distinct transition states. The authors assess this endpoint through functional assays and through comparison with bona fide neurons isolated from animals.

Essential revisions:

All three reviewers are enthusiastic about the general concept and approach of your study. However, two Reviewers note that the induction of forebrain markers in the direct protocol might be an artifact of culture conditions rather than an obligate intermediate in the pathway. The reviewers suggest that 1) knockdown of the patterning factors in the direct and standard protocol or 2) isolation of intermediates to follow their fate prospectively or 3) early induction of factors by dox would strengthen the conclusions of the study and provide more mechanistic insight. Two reviewers also note that the efficiency of induction of motor neurons in the standard protocol is very low. Repetition of some of the key experiments with a higher yield would overcome these concerns. Finally, the reviewers request more clarity around the definition of particular cell types. Specific points are listed below.

1) Though the overall concept behind this study is interesting, there are some significant issues with the experimental design. First, it is not clear to what extent the authors' observations are a function of the differences in the particular culture conditions employed-in particular transfer of pluripotent cells to serum-free medium in adherent culture in the direct protocol (which can induce forebrain specification as a default), versus three-dimensional culture or EB formation in the standard protocol (which can markedly influence heterogeneity and outcomes). Perhaps a greater concern with this study is the very low yields of the desired cell type from the standard protocol. The motor neuron yields peak at Day 5 with the standard protocol (about 25% early motor neurons) then a variety of other cell types are generated such that by Day 12, there are very few cells with properties of motor neurons. By contrast the direct protocol is highly efficient (62.7% versus 2.7% motor neurons). It is questionable whether any meaningful comparison can be drawn between these two protocols; when one yields only 3% of the desired endpoint using a different culture platform, it is hardly surprising to observe different intermediates. The protocol the authors follow for standard differentiation (Wu et al., 2012) reports ~50% motor neuron yield at 7 days; much closer to the direct protocol yields the authors report; it is not clear why they did not achieve a better yield in this study from the protocol of Wu et al., 2012.

2) The percentage of MNs in the SP protocol is surprisingly low. The culture conditions, crucial for this conclusion, should be clearly described in methods. Recapitulating development, SD protocol produces oligodendrocytes after MNs, thus efficiency as percentage of total cells it is expected to decrease.

3) MN are patterned along the rostro-caudal (R-C) axis of the spinal cord to innervate the different target by Hox genes. Thus, MN subtype identity varies greatly along the R-C axis. It is expected that neither of the in vitro differentiation protocol decapitates the diversity of embryonic derived MNs. Specifically, the SP was designed to make a very restricted set of MNs within the R-C axis. Without description of the method, it is left to the reader to assume MNs were collected from all R-C levels.

4) Single cell RNA seq data is biased towards detecting highly expressed genes. While the DP protocol appears to follow a very artificial trajectory it is possibly that key genes that are crucial for generation of the SD intermediates that the DP approach seems to skip are transiently expressed at very low levels. The authors could/should show that shRNA mediated knockdown of the key TF that are important for in vivo/SD generation of MNs but appear to get skipped in the DP protocol, reduce differentiation outcomes for the SD but not the DP protocol. This will be crucial to support that DP MNs are indeed derived by a process that skips/bypasses key developmental intermediates.

5) It will be important to investigate if the generation of Sox1/Sox3+ neural progenitors during the DP protocol is not simply an artefact of the authors experimental conditions. The authors cited (3) and stated in the introduction that MNs are generated by the DP approach by skipping the a Sox1 positive neural progenitor state. As neural commitment in N2B27 is a very rapid process and mESCs are exposed to the dox and N2B27 media at the same time it is possible that the cells start to differentiate before the NIL factors become active on a protein level. While the authors state that MNs cannot be generated by dox addition if cells are not removed from mESC maintenance media, pulsing the cells for 12, 24, or 48 hours with dox in mESC media before on set of differentiation very well might (the original DR publication shows that NIL factors are present on protein level by 48 hours). I understand that asking the authors to repeat the single cell transcriptome under this condition will be excessive, however I propose that the authors use qPCR (single cell) for neural progenitor markers and mature MN markers to test whether pre-exposure to dox abolishes the generation of neural progenitors during N2B27 culture. If it does this would indicate that SD/DR approaches are entirely different processes that do not even share the neural progenitor state and more importantly that mESCs do directly transition to MN-like state.

6) As commented above, it would be interesting to test whether knock-down of forebrain transcription factors that get transiently upregulated during DP mediated MN generation reduce differentiation outcomes for the DP protocol. This would help to demonstrate more clearly and in a mechanistic way, whether the forebrain signature is important for the process or merely an artefact imposed by the N2B27 media which generates under normal differentiation conditions forebrain fates.

7) Finally, to definitely proof that both the SD and DP protocol give rise to MN via different intermediate stages, it will be critical to isolate the presumed intermediate cells by FACS (either by using specific cell surface markers inferred from the single cell data or at least they should use their MNX1:GFP line) and show that these "presumed" intermediates are indeed the source of the mature MNs. That is, the authors should re-culture the isolated cells and demonstrate by IF characterisation that they give rise to the final end product.

Reviewer #1:

This manuscript uses single cell transcriptomics to compare the cell state transitions that occur when mouse embryonic stem cells are directly reprogrammed into motor neurons using defined transcription factors (direct protocol) to those that occur during a directed differentiation designed to simulate normal embryological induction processes (standard protocol). The main conclusion is that the two pathways converge on a similar endpoint via distinct transition states. The authors assess this endpoint through functional assays and through comparison with bona fide neurons isolated from animals.

The manuscript addresses an important question in cellular differentiation and stem cell biology. It is fair to point out that the direct reprogramming of embryonic stem cells (which are pluripotent) is liable to be significantly different in important ways to direct reprogramming of differentiated adult somatic cells (the more likely starting point for many direct reprogramming protocols). The latter type of conversion most often requires crossing or upwards traverse of cell lineages whereas the former might be expected to be more closely related to normal developmental pathways. All the same this study is certainly a valid model for direct reprogramming, and having a common starting point for the standard and direct protocols simplifies the analysis.

Though the overall concept behind this study is interesting, there are some significant issues with the experimental design. First, it is not clear to what extent the authors' observations are a function of the differences in the particular culture conditions employed-in particular transfer of pluripotent cells to serum-free medium in adherent culture in the direct protocol (which can induce forebrain specification as a default), versus three-dimensional culture or EB formation in the standard protocol (which can markedly influence heterogeneity and outcomes). Perhaps a greater concern with this study is the very low yields of the desired cell type from the standard protocol. The motor neuron yields peak at Day 5 with the standard protocol (about 25% early motor neurons) then a variety of other cell types are generated such that by Day 12, there are very few cells with properties of motor neurons. By contrast the direct protocol is highly efficient (62.7% versus 2.7% motor neurons). It is questionable whether any meaningful comparison can be drawn between these two protocols; when one yields only 3% of the desired endpoint using a different culture platform, it is hardly surprising to observe different intermediates. The protocol the authors follow for standard differentiation (Wu et al., 2012) reports ~50% motor neuron yield at 7 days; much closer to the direct protocol yields the authors report; it is not clear why they did not achieve a better yield in this study from the protocol of Wu et al., 2012.

1) Results section–sometimes embryos do reach a common terminal differentiation endpoint through different pathways. Cartilage (derived from neural crest or mesoderm) is one example and there are others.

2) Results section–be more explicit about how the various cell states are defined.

3) Results section–how exactly were these efficiencies calculated? Do they take into account cell proliferation and loss at various stages?

4) Results section–is the intermediate DP state a necessary stage on the journey? Maybe it represents a default (early anterior stages) induced by growth factor depletion that then gets overtaken by the reprogramming factors.

5) Discussion section– because the MN endpoint is not characterised at a functional level by the authors in this study, the conclusions rely on comparison to historical controls, and therefore depend on the degree to which the authors' protocol mimics earlier efforts. Direct comparison of the two endpoints is only possible at the transcriptional level.

Reviewer #2:

Briggs, Li and coworkers aim to investigate if the developmental path taken by differentiating cells determines the final cellular state or if two alternative trajectories are able to produce a similar terminal fate. To do so, the collect single cell RNA-seq data from differentiating motor neurons induced by extracellular signals or by forced expression of transcription factors. An important difference between these samples rests on the fact that direct differentiation by forced transcription factor expression bypasses the canonical motor neuron progenitor stages. The results show that direct programming and standard differentiating cells diverge at early progenitor like state to take different path and converge on a final similar cell state. Although this was postulated before, these results test the hypothesis with the appropriate and up to date technology.

There are some important points that in my opinion should be addressed;

1) A better description of the criteria used to assign cell types will be useful. For example, motor neuron progenitors (MNP) are characterized by Olig2 expression, low or no Isl1, Mnx1 and Lhx3. In Figure 1, MNP have an average Olig2 and high Mnx1, Isl1 and Lhx3. Thus, some of the cells in this cluster might be early MNs than progenitors. Are the markers in Figure 1 some of the genes driving the clustering?

2) The percentage of MNs in the SP protocol is surprisingly low. The culture conditions, crucial for this conclusion, should be clearly described in methods. Recapitulating development, SD protocol produces oligodendrocytes after MNs, thus efficiency as percentage of total cells it is expected to decrease. Where there antimitotic added to both (DP and SP) conditions to restrict the generation of mitotic byproducts?

3) It is my opinion that Figure 5 should be divided into two figures. The first two panels are intrinsically linked to the goal of this manuscript, the characterization of DP MNs is less relevant and novel. The three-way comparison of SP, DP and PM is extremely valuable and on topic. Thus, I believe I deserves to be expanded and explained better. The supplementary figures associate with this figure should be in the main figure. On that note, it will be useful to color Figure 5—figure supplement 2A based on DP, SP and pMN to understand where cells "land" in these dimensions. Describe what is gene set that take DP closer to pMN than SP for example? Is it possible to add pMN and remake here Figure 3?

4) MN are pattern along the rostro-caudal (R-C) axis of the spinal cord to innervate the different target by Hox genes. Thus, MN subtype identity varies greatly along the R-C axis. It is expected that neither of the in vitro differentiation protocol decapitates the diversity of embryonic derived MNs. Specifically, the SP was designed to make a very restricted set of MNs within the R-C axis. Without description of the method, it is left to the reader to assume MNs were collected from all R-C levels. Thus, it this is not a surprised or anything that should be left as a speculation and stated that way.

5) Although with some novel detail, the rest of Figure 5 is a recapitulation of previous work. Finally, the Tubb3+BTX staining is not informative. Also, clustering of acetylcholine receptors when cells are platted without neurons is required to conclude that MNs induce clustering. A Vacht or SV2 staining plus BTX and quantification of the overlap will be a better measure of active NMJs.

6) The finding that in the DP protocol cells don't transition through MNP stages is not novel. However, I do understand that for the purpose of comparing DP to SP it is a necessary description. Single cell RNA-seq and differentiation path is the appropriate and cutting edge tool to investigate the original question: does developmental path matters? The cellular system is appropriate too since three types of MNs can be compared: DP, SP, and pMN. The manuscript is clearly written following a logical argument. I believe it suffers from a very poor description of the experiment and methods that hurst the interpretation of the data. Also, it is superficial in important description as those single cell RNA-seq in Figure 5.

Reviewer #3:

The study, "Mouse embryonic stem cells can differentiate via multiple paths to the same state" by Briggs et al. uses single cell sequencing to demonstrate that mouse ESCs can give rise to motor neurons via alternative transcriptional trajectories. In detail the authors compare a traditional cytokine driven differentiation protocol (SD) with a transcription factor based direct "programming" approach. While the authors were able to detect discreet developmental intermediates in the SD protocol that correspond to known in vivo counter parts, the DP approach followed a mostly alternative path. While both DP and SD cultures appeared to transition into neural progenitors initially, beyond that they seemed to directly transition into an early motor neuron state bypassing a number of intermediate stages. The authors postulate that the same cell fate can be derived from mESCs via alternative routes that diverge after the neural commitment stage.

This is an important and highly interesting study that tries to unveil how direct programming produces its target cell type and whether this is achieved by an accelerated form of differentiation (transcription factor guided) or whether this occurs via an alternative route/more direct cell state transition. While the authors use state of the art bioinformatical analyses to mine their data set is well done, and allowed them reach interesting results, I believe that a few key experiments are required for the authors to mechanistically support their key findings.

– Single cell RNA seq data is biased towards detecting highly expressed genes. While the DP protocol appears to follow a very artificial trajectory it is possibly that key genes that are crucial for generation of the SD intermediates that the DP approach seems to skip are transiently expressed at very low levels. The authors could/should show that shRNA mediated knockdown of the key TF that are important for in vivo/SD generation of MNs but appear to get skipped in the DP protocol, reduce differentiation outcomes for the SD but not the DP protocol. This will be crucial to support that DP MNs are indeed derived by a process that skips/bypasses key developmental intermediates.

– It will be important to investigate if the generation of Sox1/Sox3+ neural progenitors during the DP protocol is not simply an artefact of the authors experimental conditions. The authors cited (3) and stated in the introduction that MNs are generated by the DP approach by skipping the a Sox1 positive neural progenitor state. As neural commitment in N2B27 is a very rapid process and mESCs are exposed to the dox and N2B27 media at the same time it is possible that the cells start to differentiate before the NIL factors become active on a protein level. While the authors state that MNs can not be generated by dox addition if cells are not removed from mESC maintenance media, pulsing the cells for 12, 24, or 48 hours with dox in mESC media before on set of differentiation very well might (the original DR publication shows that NIL factors are present on protein level by 48 hours). I understand that asking the authors to repeat the single cell transcriptome under this condition will be excessive, however I propose that the authors use qPCR (single cell) for neural progenitor markers and mature MN markers to test whether pre-exposure to dox abolishes the generation of neural progenitors during N2B27 culture. If it does this would indicate that SD/DR approaches are entirely different processes that do not even share the neural progenitor state and more importantly that mESCs do directly transition to MN-like state.

– As commented above, it would be interesting to test whether knock-down of forebrain transcription factors that get transiently upregulated during DP mediated MN generation reduce differentiation outcomes for the DP protocol. This would help to demonstrate more clearly and in a mechanistic way, whether the forebrain signature is important for the process or merely an artefact imposed by the N2B27 media which generates under normal differentiation conditions forebrain fates.

– Finally, to definitely proof that both the SD and DP protocol give rise to MN via different intermediate stages, it will be critical to isolate the presumed intermediate cells by FACS (either by using specific cell surface markers inferred from the single cell data or at least they should use their MNX1:GFP line) and show that these "presumed" intermediates are indeed the source of the mature MNs. That is, the authors should re-culture the isolated cells and demonstrate by IF characterisation that they give rise to the final end product.

An experiment that I have not suggested because it maybe an excess, but it would really enhance this manuscript, would be to use lineage tracing for Olig2 or Nkx6-1 during the DP protocol, and in this way show that they are never upregulated during the generation of MN by the DP approach.

---

## [Author Response]

Essential revisions:All three reviewers are enthusiastic about the general concept and approach of your study. However, two Reviewers note that the induction of forebrain markers in the direct protocol might be an artifact of culture conditions rather than an obligate intermediate in the pathway. The reviewers suggest that 1) knockdown of the patterning factors in the direct and standard protocol or 2) isolation of intermediates to follow their fate prospectively or 3) early induction of factors by dox would strengthen the conclusions of the study and provide more mechanistic insight.

We thank the reviewers and editors for their enthusiasm for our paper. Of the three options suggested for extended validation of our original study stated above, we chose to follow option 2). Specifically, we isolated the DP transitional state by FACS using a Mnx1:GFP reporter cell line. We then prospectively cultured this population, and validated by immunostaining that it indeed generates motor neurons. This experiment confirms that the abnormal transitional state in DP gives rise to MNs.

Two reviewers also note that the efficiency of induction of motor neurons in the standard protocol is very low. Repetition of some of the key experiments with a higher yield would overcome these concerns.

Here, the major concern was that our MN production efficiency of 2.7% is extremely low compared to the reported efficiency of ~50%, e.g. by Wu et al., 2012. This raised a legitimate concern about whether any of the further analysis in the paper could be relied on, if one of the protocols simply failed. We have addressed this point, both through correcting our analysis and by collecting additional data.

For the former, we realized that the discrepancy between 2.7% and 50% arises almost entirely because we used a much more stringent definition for an MN than the “classical” method for scoring MN efficiency, defined by Wu et al., 2012. Wu et al. identified motor neurons as any GFP+ cells in an Mnx1:GFP reporter system. Since GFP is a stable molecule, and Mnx1 is first expressed in only motor neuron progenitors, the “classical” definition of efficiency includes everything from the earliest committed motor neuron progenitors through to the most mature motor neuron state present. Wu et al.’s method is suitable when analyzing a culture that is well differentiated. However, it cannot determine whether the GFP+ cells are mature MNs.

Crucially, when we retrospectively apply the same single biomarker criterion, we identify motor neurons produced at an efficiency of 42.2% for the standard protocol. This number specifically reflects the number of GFP+ cells using an Mnx1:GFP reporter system, as in Wu et al., 2012.

The power of scRNA-Seq is precisely in that it can use more than a single biomarker in defining cell states. Our scRNA-Seq-based criteria was far more stringent, and referred to the mature subset of what is included in Wu et al.’s definition. Overall, this emphasizes a major advantage of using scRNA-Seq to study in vitro differentiation. To avoid confusion, in the revised text we begin our discussion of differentiation efficiency with the widely-used Mnx1+ efficiency estimate, before moving on to more refined estimates using our scRNA-seq data.

Also, as requested, we performed a replicate SP experiment and profiled an additional 1,372 cells by InDrops. This time, we analyzed cells from day 10 of differentiation, at which time we expect the efficiency of EMN and LMN states to be maximal. At this time point, we found that 49% of all cells (n=675) were EMN or LMN. This result demonstrates that the method works efficiently in our hands. The results are now included in (Figure 3—figure supplement 1, Supplementary Materials and methods). We have also integrated these additional data into all of our analyses comparing the DP and SP trajectories, thereby gaining additional statistical power.

Overall, this analysis makes the paper much clearer and stronger than in our first version, and we are grateful to the reviewers for raising this point.

Finally, the reviewers request more clarity around the definition of particular cell types. Specific points are listed below.1) Though the overall concept behind this study is interesting, there are some significant issues with the experimental design. First, it is not clear to what extent the authors' observations are a function of the differences in the particular culture conditions employed-in particular transfer of pluripotent cells to serum-free medium in adherent culture in the direct protocol (which can induce forebrain specification as a default), versus three-dimensional culture or EB formation in the standard protocol (which can markedly influence heterogeneity and outcomes). Perhaps a greater concern with this study is the very low yields of the desired cell type from the standard protocol. The motor neuron yields peak at Day 5 with the standard protocol (about 25% early motor neurons) then a variety of other cell types are generated such that by Day 12, there are very few cells with properties of motor neurons.

This particular comment is due to the definition of what is normally scored as MN, and it is now addressed by our response above.

By contrast the direct protocol is highly efficient (62.7% versus 2.7% motor neurons). It is questionable whether any meaningful comparison can be drawn between these two protocols; when one yields only 3% of the desired endpoint using a different culture platform, it is hardly surprising to observe different intermediates. The protocol the authors follow for standard differentiation (Wu et al., 2012) reports ~50% motor neuron yield at 7 days; much closer to the direct protocol yields the authors report; it is not clear why they did not achieve a better yield in this study from the protocol of Wu et al., 2012.

There appear to be two major concerns in this comment. (1) Our observation of alternative differentiation paths might be due to trivial differences between the two protocols; and (2) one of the protocols give very low yields compared to its reference publication (Wu et al). We addressed the concern (2) directly above, showing that our results are in agreement with Wu et al., 2012. Now for concern (1):

Rather than being an accidental issue of poor experimental design, the different nature of the protocols we chose to compare was in fact by design. This may not have been clear in the text previously. We clarify these goals now in the Introduction of the revised manuscript.

We chose two protocols that were very different in the hopes of identifying alternate trajectories, and asking whether the end points reached are nonetheless the same. This was not guaranteed. Over a dozen alternative motor neuron protocols have been demonstrated in the literature, each variously using 2D or 3D culture, serum or serum-free culture conditions, and different regimes or growth factor additions. Nonetheless these protocols are all understood to drive cells through similar and recognizable intermediate states on the way to becoming a motor neuron (see for example Figure 2 of Sances et al., 2016). Indeed DP and SP share their first intermediate state in our data, illustrating how very different culture environments can shuttle cells through similar states. DP (as originally reported by Mazzoni et al., 2013) was among the first methods for which it was explicitly argued that a distinct process may be occurring. Our focus in this manuscript was to discover what this new process was in comparison to an example of a method that drives cells through an embryo-like sequence of states. The question as to the contributions to the differences (media or otherwise) comes secondary to the main phenomenon.

2) The percentage of MNs in the SP protocol is surprisingly low.

We discussed the seemingly low SP efficiency above.

The culture conditions, crucial for this conclusion, should be clearly described in methods.

We include more detailed descriptions of our differentiation methods in subsection “Dissection of two MN differentiation protocols using single cell RNA sequencing” of the revised manuscript, and in the revised Materials and methods section.

Recapitulating development, SD protocol produces oligodendrocytes after MNs, thus efficiency as percentage of total cells it is expected to decrease.

The reviewer also correctly notes that SP produces off-target populations such as oligodendrocytes after MNs, diluting total efficiency over time. We make this point now in subsection “Dynamics of MN generation versus off-target differentiation in DP and the SP” of the revised manuscript.

3) MN are patterned along the rostro-caudal (R-C) axis of the spinal cord to innervate the different target by Hox genes. Thus, MN subtype identity varies greatly along the R-C axis. It is expected that neither of the in vitro differentiation protocol decapitates the diversity of embryonic derived MNs. Specifically, the SP was designed to make a very restricted set of MNs within the R-C axis. Without description of the method, it is left to the reader to assume MNs were collected from all R-C levels.

The reviewer correctly points out that the SP was designed to make motor neurons of spinal identity. Similarly, DP was also designed to generate spinal motor neurons. We now clarify these details in subsection “Dissection of two MN differentiation protocols using single cell RNA sequencing” of the manuscript. We also clarify that E13.5 primary motor neurons were collected from whole spinal cords subsection “Both DP and the SP approach a transcriptional state similar to *bona-fide* MNs in embryos”.

4) Single cell RNA seq data is biased towards detecting highly expressed genes. While the DP protocol appears to follow a very artificial trajectory it is possibly that key genes that are crucial for generation of the SD intermediates that the DP approach seems to skip are transiently expressed at very low levels. The authors could/should show that shRNA mediated knockdown of the key TF that are important for in vivo/SD generation of MNs but appear to get skipped in the DP protocol, reduce differentiation outcomes for the SD but not the DP protocol. This will be crucial to support that DP MNs are indeed derived by a process that skips/bypasses key developmental intermediates.

The concern raised here is that key genes of the SD intermediates are expressed in the DP transition and could be required for fate specification. We decided to pursue an alternative option to shRNA-based perturbation, for two reasons:

Reason (1): we were concerned that efficient shRNA knockdown can require optimization, and that variable delivery of shRNA constructs across a population of cells may introduce its own artifacts, e.g. leading to non-specific fate perturbations easily detected by scRNA-Seq. Ultimately these non-specific effects could be controlled for through controls, but with a short response time this seemed risky and likely to generate more artifacts than answers. The turn-around time and cost of scRNA-Seq for each round of shRNA would have been prohibitive for the 2 month response time requested. Our caution here was supported by the editor’s suggestion, above, that we need not pursue all three major experimental demands.

Reason (2): it seems that the underlying concern raised here is about whether these genes are expressed at all, since if their expression can be proven to be negligible, then so it their effect. We conducted new qPCR experiments, and further statistical analysis of single cell data, to argue that the key SP genes are reliably not detectable at functional levels in the DP protocol:

– New qPCR measurements show that the levels of Olig2 in DP are 10 times lower than we would expect if just 0.1% of the DP population expressed just 1 molecule per cell (Figure 2—figure supplement 3).

– Consistent with this, in our scRNA-Seq data, we estimate the expression level of Olig2 and Nkx6-1 level to be < 0.5 molecules per cell in DP intermediate states, compared to 172 and 38 molecules per cell respectively in SP intermediates. The large number of cells that we profiled ensure that the associated error in these estimates is low.

– We now ask whether rare Olig2+ cells may exist as a hidden subpopulation within our data that our clustering approach simply failed to resolve. Taking our qPCR data as a bound on Olig2 expression across the population we calculate that even if the hidden population expressed just 1 molecule per cell, and represented 0.1% of the total population, it must have a lifetime on <15 minutes. This is inconsistent with the lifetime of mRNA molecules inside and so we conclude that the hidden subpopulation must not exist.

Together these results argue with high sensitivity that Olig2 and Nkx6-1, two key SP intermediate genes, are not expressed even transiently at comparable levels during DP. These analyses are provided now in detail in subsection “The DP differentiation trajectory lacks intermediates expressing Olig2 and Nkx6-1” of the revised manuscript. This falls short of the definitive rigor of a well-controlled gene ablation experiment, but is nonetheless extremely strong quantitative evidence.

5) It will be important to investigate if the generation of Sox1/Sox3+ neural progenitors during the DP protocol is not simply an artefact of the authors experimental conditions. The authors cited (3) and stated in the introduction that MNs are generated by the DP approach by skipping the a Sox1 positive neural progenitor state. As neural commitment in N2B27 is a very rapid process and mESCs are exposed to the dox and N2B27 media at the same time it is possible that the cells start to differentiate before the NIL factors become active on a protein level. While the authors state that MNs cannot be generated by dox addition if cells are not removed from mESC maintenance media, pulsing the cells for 12, 24, or 48 hours with dox in mESC media before on set of differentiation very well might (the original DR publication shows that NIL factors are present on protein level by 48 hours). I understand that asking the authors to repeat the single cell transcriptome under this condition will be excessive, however I propose that the authors use qPCR (single cell) for neural progenitor markers and mature MN markers to test whether pre-exposure to dox abolishes the generation of neural progenitors during N2B27 culture. If it does this would indicate that SD/DR approaches are entirely different processes that do not even share the neural progenitor state and more importantly that mESCs do directly transition to MN-like state.

The experiment proposed in this comment is very interesting, and mirrors some of our own thinking about future work, as laid out in the manuscript Discussion section (paragraph 3). We would prefer it to leave it as a future direction as it does not directly bear on the validity or interest of our manuscripts central conclusion – that mESC differentiation can proceed via multiple paths to the same state.

More specifically: If the proposed experiment turns out as anticipated, it would show that the early Sox1+ state identified in both DP and SP protocols could be also skipped in DP protocol under a new condition of DOX pre-treatment. This would establish that a third differentiation pathway exists, in addition to the two we already describe. Conversely, if DOX pre-treatment failed to change the dynamics, it would still remain true that two pathways exist to the same end state. It is indeed fascinating to know how malleable these differentiation trajectories truly are. We think that the idea of testing the limits of plasticity of differentiation trajectories is fascinating, can be addressed even more generally than proposed here, and it is worthy of a complete and separate investigation. As above, our caution here was supported by the editor’s suggestion, above, that we need not pursue all three major experimental demands.

6) As commented above, it would be interesting to test whether knock-down of forebrain transcription factors that get transiently upregulated during DP mediated MN generation reduce differentiation outcomes for the DP protocol. This would help to demonstrate more clearly and in a mechanistic way, whether the forebrain signature is important for the process or merely an artefact imposed by the N2B27 media which generates under normal differentiation conditions forebrain fates.

While we agree that this experiment could in principle be interesting, similarly to our previous response we feel that this next step of interrogation of one specific mechanistic hypothesis goes beyond the scope of our present study. In particular: if the forebrain genes do turn out to be essential for DP, it would add some molecular dependencies to the new cell state transitions that we see. Alternatively, if the forebrain genes are not essential to DP, we would learn that not all of the genes activated during one differentiation path from mESCs to MNs are necessary, which would not be too surprising. In either case, our demonstration in this paper that multiple paths to the same state exist would be unchanged. Moreover, whether or not forebrain gene expression is necessary for DP differentiation, it remains interesting to observe that forebrain expression is at least compatible with MN generation, as this is never observed in embryos. It is suggestive of how conserved cell states may be generated in new body locations during evolution, and firmly debunks the view that typical spinal intermediate states are necessary to pass down a MN lineage.

7) Finally, to definitely proof that both the SD and DP protocol give rise to MN via different intermediate stages, it will be critical to isolate the presumed intermediate cells by FACS (either by using specific cell surface markers inferred from the single cell data or at least they should use their MNX1:GFP line) and show that these "presumed" intermediates are indeed the source of the mature MNs. That is, the authors should re-culture the isolated cells and demonstrate by IF characterisation that they give rise to the final end product.

We thank the reviewer for this suggestion, which we have now carried out. In the revised manuscript submission we include exactly this experiment – see revised Figure 4. By using the Mnx1:GFP reporter cell line, we isolated the DP transitional state cells on day 3 of differentiation, cultured them for an additional 10 days, and show by immunostaining that they indeed generate mature MNs (Vacht/Map2+).

Reviewer #1:This manuscript uses single cell transcriptomics to compare the cell state transitions that occur when mouse embryonic stem cells are directly reprogrammed into motor neurons using defined transcription factors (direct protocol) to those that occur during a directed differentiation designed to simulate normal embryological induction processes (standard protocol). The main conclusion is that the two pathways converge on a similar endpoint via distinct transition states. The authors assess this endpoint through functional assays and through comparison with bona fide neurons isolated from animals.The manuscript addresses an important question in cellular differentiation and stem cell biology. It is fair to point out that the direct reprogramming of embryonic stem cells (which are pluripotent) is liable to be significantly different in important ways to direct reprogramming of differentiated adult somatic cells (the more likely starting point for many direct reprogramming protocols). The latter type of conversion most often requires crossing or upwards traverse of cell lineages whereas the former might be expected to be more closely related to normal developmental pathways. All the same this study is certainly a valid model for direct reprogramming, and having a common starting point for the standard and direct protocols simplifies the analysis.Though the overall concept behind this study is interesting, there are some significant issues with the experimental design. First, it is not clear to what extent the authors' observations are a function of the differences in the particular culture conditions employed-in particular transfer of pluripotent cells to serum-free medium in adherent culture in the direct protocol (which can induce forebrain specification as a default), versus three-dimensional culture or EB formation in the standard protocol (which can markedly influence heterogeneity and outcomes). Perhaps a greater concern with this study is the very low yields of the desired cell type from the standard protocol. The motor neuron yields peak at Day 5 with the standard protocol (about 25% early motor neurons) then a variety of other cell types are generated such that by Day 12, there are very few cells with properties of motor neurons. By contrast the direct protocol is highly efficient (62.7% versus 2.7% motor neurons). It is questionable whether any meaningful comparison can be drawn between these two protocols; when one yields only 3% of the desired endpoint using a different culture platform, it is hardly surprising to observe different intermediates. The protocol the authors follow for standard differentiation (Wu et al., 2012) reports ~50% motor neuron yield at 7 days; much closer to the direct protocol yields the authors report; it is not clear why they did not achieve a better yield in this study from the protocol of Wu et al., 2012.

This comment is addressed above in the editor’s “essential revisions” comments.

1) Results section–sometimes embryos do reach a common terminal differentiation endpoint through different pathways. Cartilage (derived from neural crest or mesoderm) is one example and there are others.

This is a fair point, and we thank the reviewer for the clarification. We have modified the referenced sentence in the Introduction to now read: “Furthermore, most mature cell states are not produced by multiple differentiation paths in the embryo; the only known exceptions are rare and viewed as special cases, such as in the neural crest lineage.”

2) Results section–be more explicit about how the various cell states are defined

This comment is addressed above in the editor’s “essential revisions” comments.

3) Results section–how exactly were these efficiencies calculated? Do they take into account cell proliferation and loss at various stages?

This comment is addressed above in the editor’s “essential revisions” comments.

4) Results section–is the intermediate DP state a necessary stage on the journey? Maybe it represents a default (early anterior stages) induced by growth factor depletion that then gets overtaken by the reprogramming factors

Please see above responses to essential revisions points 5, 6, and 7. While we do not prove that the novel DP transitional state is necessary for MN production in DP, we do now include additional experiments showing that this abnormal state can be isolated and that it has MN generation potential. In our view establishing a clear view of a single instance of differentiation through alternate pathways is in itself interesting, and its demonstration set the scope of this work. The plasticity of the intermediate state and the precise origin of the dynamics are questions we would leave for future work

5) Discussion section– because the MN endpoint is not characterised at a functional level by the authors in this study, the conclusions rely on comparison to historical controls, and therefore depend on the degree to which the authors' protocol mimics earlier efforts. Direct comparison of the two endpoints is only possible at the transcriptional level.

The MN endpoint is functionally interrogated for DP MNs in Figure 5 (Figure 6 in the revised manuscript). We assume the reviewer therefore refers to SP MNs in this comment. In response we would like to clarify that SP has been interrogated by the same assays by a number of labs, including by one the authors (Clifford Woolf), whose lab has supervised our implementation of SP, and confirmed that it behaves typically. Our functional results for DP MNs are within the range of values typically seen for SP MNs.

Reviewer #2:Briggs, Li and coworkers aim to investigate if the developmental path take by differentiating cells determines the final cellular state or if two alternative trajectories are able to produce a similar terminal fate. To do so, the collect single cell RNA-seq data from differentiating motor neurons induced by extracellular signals or by forced expression of transcription factors. An important difference between these samples rests on the fact that direct differentiation by forced transcription factor expression bypasses the canonical motor neuron progenitor stages. The results show that direct programming and standard differentiating cells diverge at early progenitor like state to take different path and converge on a final similar cell state. Although this was postulated before, these results test the hypothesis with the appropriate and up to date technology.There are some important points that in my opinion should be addressed;1) A better description of the criteria used to assign cell types will be useful. For example, motor neuron progenitors (MNP) are characterized by Olig2 expression, low or no Isl1, Mnx1 and Lhx3. In Figure 1, MNP have an average Olig2 and high Mnx1, Isl1 and Lhx3. Thus, some of the cells in this cluster might be early MNs than progenitors. Are the markers in Figure 1 some of the genes driving the clustering?

This comment is addressed above in the editor’s “essential revisions” comments.

2) The percentage of MNs in the SP protocol is surprisingly low. The culture conditions, crucial for this conclusion, should be clearly described in methods. Recapitulating development, SD protocol produces oligodendrocytes after MNs, thus efficiency as percentage of total cells it is expected to decrease. Where there antimitotic added to both (DP and SP) conditions to restrict the generation of mitotic byproducts?

This comment is addressed above in the editor’s “essential revisions” comments. We have included additional descriptions of our culture conditions in the revised methods, and discussed how ongoing proliferation may dilute post-mitotic MNs in SP in the revised main text.

3) It is my opinion that Figure 5 should be divided into two figures. The first two panels are intrinsically linked to the goal of this manuscript, the characterization of DP MNs is less relevant and novel. The three-way comparison of SP, DP and PM is extremely valuable and on topic. Thus, I believe I deserves to be expanded and explained better. The supplementary figures associate with this figure should be in the main figure. On that note, it will be useful to color Figure 5—figure supplement 2A based on DP, SP and pMN to understand where cells "land" in these dimensions. Describe what is gene set that take DP closer to pMN than SP for example? Is it possible to add pMN and remake here Figure 3?

We have followed this recommendation and split the previous Figure 5 into two figures. In the revised manuscript Figure 5 expands upon the previous comparisons between the DP and SP terminal states and pMNs, including elements the previous supplementary material as well as novel analyses that dig deeper into the similarities and differences between states. Figure 6 is now a self-contained figure with all of the molecular and functional characterization of the DP terminal state.

4) MN are pattern along the rostro-caudal (R-C) axis of the spinal cord to innervate the different target by Hox genes. Thus, MN subtype identity varies greatly along the R-C axis. It is expected that neither of the in vitro differentiation protocol decapitates the diversity of embryonic derived MNs. Specifically, the SP was designed to make a very restricted set of MNs within the R-C axis. Without description of the method, it is left to the reader to assume MNs were collected from all R-C levels. Thus, it this is not a surprised or anything that should be left as a speculation and stated that way.

This comment is addressed above in the editor’s “essential revisions” comments.

5) Although with some novel detail, the rest of Figure 5 is a recapitulation of previous work. Finally, the Tubb3+BTX staining is not informative. Also, clustering of acetylcholine receptors when cells are platted without neurons is required to conclude that MNs induce clustering. A Vacht or SV2 staining plus BTX and quantification of the overlap will be a better measure of active NMJs.

The validation experiments in Figure 5 (Figure 6 in the revised manuscript) are a confidence building exercise that emphasizes how the abnormal DP differentiation trajectory still makes functional MNs. Although it is not the most novel part of the study, these precise controls have not been shown previously for our exact DP culture conditions, and we believe them to be important. Regarding the previous Figure 5, the Tubb3+BTX staining is a simple accessory to the +curare muscle contraction assay; it lets the reader visualize both neurons and muscle cells in the co-culture system. The broader point of the panel is that acetylcholine-dependent muscle contraction occurs, showing that DP MNs are communicating with muscle cells. We separately include Vacht staining in Figure 4 and Figure 6 in the revised manuscript.

6) The finding that in the DP protocol cells don't transition through MNP stages is not novel. However, I do understand that for the purpose of comparing DP to SP it is a necessary description. Single cell RNA-seq and differentiation path is the appropriate and cutting edge tool to investigate the original question: does developmental path matters? The cellular system is appropriate too since three types of MNs can be compared: DP, SP, and pMN. The manuscript is clearly written following a logical argument. I believe it suffers from a very poor description of the experiment and methods that hurst the interpretation of the data. Also, it is superficial in important description as those single cell RNA-seq in Figure 5.

We have increased the amount of detail provided in the manuscript regarding our definitions of cell states and our experimental methods, which should address all of the concerns here, including the data in Figure 5.

Reviewer #3:The study, "Mouse embryonic stem cells can differentiate via multiple paths to the same state" by Briggs et al. uses single cell sequencing to demonstrate that mouse ESCs can give rise to motor neurons via alternative transcriptional trajectories. In detail the authors compare a traditional cytokine driven differentiation protocol (SD) with a transcription factor based direct "programming" approach. While the authors were able to detect discreet developmental intermediates in the SD protocol that correspond to known in vivo counter parts, the DP approach followed a mostly alternative path. While both DP and SD cultures appeared to transition into neural progenitors initially, beyond that they seemed to directly transition into an early motor neuron state bypassing a number of intermediate stages. The authors postulate that the same cell fate can be derived from mESCs via alternative routes that diverge after the neural commitment stage.This is an important and highly interesting study that tries to unveil how direct programming produces its target cell type and whether this is achieved by an accelerated form of differentiation (transcription factor guided) or whether this occurs via an alternative route/more direct cell state transition. While the authors use state of the art bioinformatical analyses to mine their data set is well done, and allowed them reach interesting results, I believe that a few key experiments are required for the authors to mechanistically support their key findings.– Single cell RNA seq data is biased towards detecting highly expressed genes. While the DP protocol appears to follow a very artificial trajectory it is possibly that key genes that are crucial for generation of the SD intermediates that the DP approach seems to skip are transiently expressed at very low levels. The authors could/should show that shRNA mediated knockdown of the key TF that are important for in vivo/SD generation of MNs but appear to get skipped in the DP protocol, reduce differentiation outcomes for the SD but not the DP protocol. This will be crucial to support that DP MNs are indeed derived by a process that skips/bypasses key developmental intermediates.

This comment is addressed above in the editor’s “essential revisions” comments.

– It will be important to investigate if the generation of Sox1/Sox3+ neural progenitors during the DP protocol is not simply an artefact of the authors experimental conditions. The authors cited (3) and stated in the introduction that MNs are generated by the DP approach by skipping the a Sox1 positive neural progenitor state. As neural commitment in N2B27 is a very rapid process and mESCs are exposed to the dox and N2B27 media at the same time it is possible that the cells start to differentiate before the NIL factors become active on a protein level. While the authors state that MNs cannot be generated by dox addition if cells are not removed from mESC maintenance media, pulsing the cells for 12, 24, or 48 hours with dox in mESC media before on set of differentiation very well might (the original DR publication shows that NIL factors are present on protein level by 48 hours). I understand that asking the authors to repeat the single cell transcriptome under this condition will be excessive, however I propose that the authors use qPCR (single cell) for neural progenitor markers and mature MN markers to test whether pre-exposure to dox abolishes the generation of neural progenitors during N2B27 culture. If it does this would indicate that SD/DR approaches are entirely different processes that do not even share the neural progenitor state and more importantly that mESCs do directly transition to MN-like state.

This comment is addressed above in the editor’s “essential revisions” comments.

– As commented above, it would be interesting to test whether knock-down of forebrain transcription factors that get transiently upregulated during DP mediated MN generation reduce differentiation outcomes for the DP protocol. This would help to demonstrate more clearly and in a mechanistic way, whether the forebrain signature is important for the process or merely an artefact imposed by the N2B27 media which generates under normal differentiation conditions forebrain fates.

This comment is addressed above in the editor’s “essential revisions” comments.

– Finally, to definitely proof that both the SD and DP protocol give rise to MN via different intermediate stages, it will be critical to isolate the presumed intermediate cells by FACS (either by using specific cell surface markers inferred from the single cell data or at least they should use their MNX1:GFP line) and show that these "presumed" intermediates are indeed the source of the mature MNs. That is, the authors should re-culture the isolated cells and demonstrate by IF characterisation that they give rise to the final end product.

This comment is addressed above in the editor’s “essential revisions” comments.

An experiment that I have not suggested because it maybe an excess, but it would really enhance this manuscript, would be to use lineage tracing for Olig2 or Nkx6-1 during the DP protocol, and in this way show that they are never upregulated during the generation of MN by the DP approach.

We thank the reviewer for this good suggestion. Unfortunately lineage tracing was not feasible for us to complete within the two month revision period. However, we did include additional qPCR measurements that validate with high sensitivity that Olig2 and Nkx6-1 are not upregulated during DP.